# Concepts' Information Bottleneck Models

## Abstract

Concept Bottleneck Models (CBMs) provide a self-explanatory framework by making predictions based on concepts that humans can understand. However, they often fall short in overall performance and interpretability because they tend to let irrelevant information seep into the concept activations. To tackle concept leakage, we introduce an information-theoretic framework to CBMs by incorporating the Information Bottleneck (IB) principle. Our method ensures that only pertinent information is retained in the concepts by limiting the mutual information between the input data and the concepts. This shift represents a new direction for CBMs, one that not only boosts concept prediction but also reinforces the link between latent representations and comprehensible concepts, leading to a model that is both more robust and more interpretable. Our findings show that our IB-based CBMs enhance the accuracy of concept prediction and diminish concept leakage without compromising the target prediction accuracy when compared to similar models. We also introduce an innovative metric designed to evaluate the quality of concept sets by focusing on performance following interventions. This metric stands in contrast to traditional task performance measures, which can sometimes conceal the impact of concept leakage, by providing a clear and interpretable means of assessing the effectiveness of concept sets.

## 1 Introduction

Explainable AI provides transparency into complex machine learning models, making their decision-making process understandable to humans. This transparency increases trust, accountability, and the ability to identify potential biases or errors thus also increasing safety. In critical domains like healthcare and finance, where AI decisions significantly impact lives, explainability is essential for ensuring fairness and ethical alignment. It also allows for continuous model improvement by exposing flaws in training data or architecture. As AI deployment increases, explainable AI becomes a necessity for maintaining human oversight and control. We distinguish four groups of explainable models. *Post-hoc techniques* (Speith, 2022) explain trained black-box models after the fact, often by approximating their behavior with interpretable models or by providing relevant feature attributions. *Model-agnostic methods* are independent of the model's internal parameters or training process, and rely, instead, on treating the model as a black-box and analyzing its inputs and outputs. *Local interpretability methods* are a subfamily of model-agnostic methods, focusing on explaining individual predictions—examples include LIME (Ribeiro et al., 2016), GradCam (Selvaraju et al., 2020). *Global interpretability* are the second subfamily of model-agnostic methods, including Accumulated Local Effect points (Apley & Zhu, 2020) and H-statistic (Friedman & Popescu, 2008). *Self-explainable models* are ad-hoc designed and trained to be able to explain their predictions at inference time without additional models or estimations. In this work, we are focusing on the latter since such models are explainable by construction and easy to debug via altering the explanations. Thus, the self-explainable methods are positioned as promising approaches over the other existing explainable models.

Concept bottleneck models (CBMs) (Koh et al., 2020) are a self-explainable approach allowing to turn any end-to-end neural network training task into a concept-based task given the concept labels. The main desiderata behind CBMs are the ability to explain the final decision to human while operating with a set of human-understandable concepts, and the ability to take corrections on these concepts into account to re-estimate the final prediction. The advantages of such approach include higher robustness to covariate shifts and spurious correlations (should the target predictions rely only on concepts). Having a model form the final prediction based on human-understandable concepts

has one more benefit: at inference time, one could manually correct mistakes in concepts predictions therefore making target prediction more accurate.

CBMs, however, were shown to have concept leakage (Margeloiu et al., 2021; Mahinpei et al., 2021)—the phenomena of concepts activations storing more information than just the concept presence. This phenomena is an issue affecting both interpretability and intervenability. Another issue with the CBMs is their performance being lower than that of a black-box models.

Among previous work that mitigates these issues, Havasi et al.'s (2022) proposal introduced side-channel CBMs and recurrent CBMs. However, side-channel CBMs have lower intervenability, and recurrent ones break the disentanglement of concepts. Kim et al.'s (2023) work introduced probabilistic CBMs, yet it needs anchor embedding points for target prediction.

Instead, we propose a simpler way to deal with concept leakage and reduced performance without altering the architecture and without introducing a need for anchor bank. We extend the Information Bottleneck (Tishby et al., 2000) principle to the concept space to reduce the concept leakage while learning robust representations. Our main idea is to obtain concepts and representations that are maximally expressive about the labels and the concepts, respectively, while having concepts maximally compressive about the data (under marginalized representations). That is, we offer an information-theoretic approach to CBMs, what we called *Concepts' Information Bottleneck*. More specifically, we show that adding Information Bottleneck (Alemi et al., 2017; Tishby et al., 2000) to CBM training objective results in improved performance and better utilization of concepts.

The main contributions of this work are three-fold: (i) a new CBM that exploits the Information Bottleneck framework providing a significant improvement compared to both vanilla CBMs and advanced concept bottleneck models, (ii) a demonstration that CBMs while may be more compressive but throw useful information based on the lack of predictive power in comparison to an IB-regularized model, and (iii) we introduce a model-based metric to measure concept set goodness (cf. Section 4.6).

## 2 RELATED WORK

### 2.1 CONCEPT BOTTLENECK MODELS

The concept bottleneck model (Koh et al., 2020), CBM, is defined as $\hat{y} = f(g(x))$, where $x \in \mathbb{R}^D$, $g \colon \mathbb{R}^D \to \mathbb{R}^k$ is a mapping from raw feature space into the lower-dimensional concepts space, and $f \colon \mathbb{R}^K \to \mathbb{R}$ is a mapping from the concepts to the target variable. For training this model composition, a dataset of triplets $\{(x_i, c_i, y_i)\}_{i=1}^N$ is needed, where $c_{(\cdot)}$ stands for the ground-truth concepts labels which should be produced by $g$. Notice that in this setup the amount of concepts to use is fixed for a particular model and that they are trained in a supervised manner.

Intuitively, when training a CBM, one is introducing human-understandable sub-labels (concepts) which are more primitive and general than the target, and then builds a model predicting the target based solely on those explainable concepts. Training process could be organized in a three ways: (i) Independent: train $f$ using ground-truth concepts $\mathcal{C} = \{c_i\}_i$ as input, and train $g$ to predict the concepts $\mathcal{C}$. (ii) Sequential: firstly train $g$ to predict the concepts, then freeze this concept-extractor model and train $f$ on the outputs of $g$ (not on the ground truth concepts labels). (iii) Joint: Optimize the weighted sum of two loss functions simultaneously: target prediction loss and concept prediction loss: $\mathcal{L}(f(g(x)), y) + \lambda \mathcal{L}(g(x), c)$.

However, the initial setup described above has been found to have several flaws: first of all, if the concepts are soft, meaning that they can be take any value in $[0, 1]$. According to Mahinpei et al.'s (2021) findings, the model $g$ learns to incorporate more information in these continuous outputs, for instance, about PCA components of the raw data. The issue appears unrelated to the training method, as it occurs even if $g$ is trained separately from $f$, nor is it related to the selection of concepts, since leakage occurs with randomly chosen dataset divisions as concepts. Mahinpei et al. (2021) posit that even for hard concepts (each concept is clipped to $\{0, 1\}$) information may leak, though the experiments confirm it only for small-dimensional data like Deng's (2012). Secondly, Margeloiu et al. (2021) argue that the CBMs desiderata is met for independent training only: for joint and sequential a CBM learns more information about the raw data than just that presented in the concepts. Thus, concepts are not used as intended. Developing the idea of tracking concepts predictions, the authors apply saliency methods to back-trace concepts to input features and find that for neither training

method of the three derive concepts from something meaningful in the input space. Conversely, we hypothesize that by compressing the concepts and the data and, simultaneously, maximally expressing the labels and concepts through their respective variables, we could obtain better concepts and representations.

Havasi et al. (2022) introduced side-channel CBMs—the ones in which information is allowed to flow aside the concept bottleneck—and recurrent CBMs, in which the model predicts concepts one after the other, and for next concept prediction utilizes the information about previous concept predictions. However, side-channel CBMs have lower intervenability, and recurrent ones break the disentanglement of concepts. Yuksekgonul et al. (2023) proposed Post-Hoc Concept Bottleneck Models (PCBM). Such type of models utilize image embeddings from pre-trained Convolutional Neural Nets penultimate layer activations. Based on these embeddings, the authors construct a concept embeddings bank and obtain concepts predictions by either projecting a new image embedding onto those embeddings or by using a SVM trained on the bank as concepts classifier. However, these models perform well only after residual connections, similar to the ones described above, are added. This residual information flow may damage both interpretability and intervenability.

To mitigate previous limitations, Zarlenga et al. (2022) presented Concept Embedding Models (CEM)—a method bridging the gap between CBMs and black-box models via learning two vectors for each concept ("active" and "inactive"). Such approach has increased target accuracy, but requires additional regularization algorithm called 'RandInt' for CEM to be able to effectively utilize test-time interventions. Moreover, the analysis of information flow done in the paper suggests that information between inputs and concepts is monotonically increasing without any compression. The paper also introduces concepts alignment score (a metric specific for the model, more complex than just accuracy) designed to evaluate how well has CEM has learned the concepts.

Our work, unlike Zarlenga et al.'s (2022) proposal, maintains the original model concept representation space and regularizes it through our concept information bottleneck regularization. In detail, first, we incorporate mutual information constraint into loss function, thus obtaining compression of information between inputs and concept activations. Secondly, we do not utilize a pair of embeddings per concept but opt for one logit per concept, as in the original CBM setup. Finally, the novel metric we introduced measures not the quality of a model, but rather the quality of concepts sets themselves.

Kim et al. (2023) introduced ProbCBMs models, which predict a parameterized distribution of concepts (mean and standard deviation) and use anchor points for class mapping. In this work, we do not utilize these anchor points, since they increase inference costs and introduce a new hyperparameter to tune at fitting stage. We do use a variational approximation over our proposed concepts' information bottleneck to predict concepts.

## 2.2 INFORMATION BOTTLENECK

Tishby et al. (2000) introduced the information bottleneck (IB) as the minimization of the functional
$$\mathcal{L}_{\text{IB}} = I(X; Z) - \beta I(Z; Y), \tag{1}$$
where $I(\cdot; \cdot)$ is the mutual information, $\beta$ is the Lagrange multiplier, $X$, $Y$ and $Z$ are random variables that represents the data, labels, and latent representations, respectively. The motivation behind the bottleneck is to "squeeze" the relevant information about target $Y$ from $X$ into a compact representation $Z$ while minimizing the information about input $X$ in $Z$—so that the representations are free of irrelevant information from $X$. The IB's authors have also posited that good generalization is connected with memorization-compression pattern. This is the behavior in which $I(Z; Y)$ increases during the whole training time, while $I(X; Z)$ increases at first (memorization) and then decreases at later iterations (compression).

Alemi et al. (2017) extended the IB framework to deep neural networks by doing a variational approximation of latent representation $Z$. And, Kawaguchi et al. (2023) analyzed the role of IB in estimation of generalization gaps for classification task. Their result implies that by incorporating the Information Bottleneck into learning objective one may get more generalized and robust network. Unlike this previous work that studied the IB for the data and the labels, we introduced another predictive variable, the concepts, and derive an upper bound that links common predictors and the ground truth into a regularizer that enforces the memorization-compression dynamics. Moreover, we show that the concepts' information bottleneck can be used in common CBM approaches through a mutual information estimator as well.

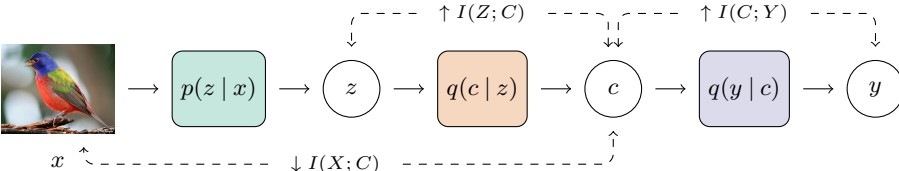

**Figure 1:** Our proposed CIBMs pipeline. The image is encoded through $p(z \mid x)$, which in turn encodes the concepts with $q(c \mid z)$, and the labels are predicted through $q(y \mid c)$. These modules are implemented as neural networks. We introduced the IB regularization as mutual information optimizations over the variables as shown in dashed lines.

## 3 CONCEPTS' INFORMATION BOTTLENECK

Concept Bottleneck Models (CBMs) aim for high interpretability by introducing human-understandable concepts, $C$, as an intermediary between latent representations, $Z$, and the labels $Y$. To preserve the interpretability at the heart of CBMs, our objective seeks to minimize $I(X;C)$—the mutual information between inputs and concepts—thereby ensuring concepts remain meaningful and free from irrelevant data, while addressing concept leakage by controlling the information flow directly at the concept level, rather than at the more abstract latent space, $Z$. Simultaneously, we aim to maximize the expressivity of the concepts about the labels, $I(C;Y)$, as well as the one of the latent representations and the concepts, $I(Z;C)$. Our initial objective is

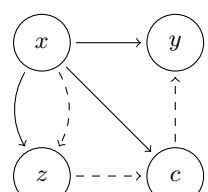

**Figure 2:** Directed graph of our model. Solid lines denote the generative model $p(y \mid x)p(c \mid x)p(z \mid x)p(x)$, and dashed lines show its variational approximation $q(y \mid c)q(c \mid z)q(z \mid x)q(x)$.

$$\max I(Z;C) + I(C;Y) \quad \text{s.t.} \quad I(X;Z) \le I_C, \qquad (2)$$

where $I_C$ is an information constraint constant, that equivalently is the maximization of the functional of the concepts' information bottleneck (CIB)

$$\mathcal{L}_{\text{CIB}} = I(Z;C) + I(C;Y) - \beta I(X;Z), \qquad (3)$$

where $\beta$ is a Lagrangian multiplier. This formulation ensures a strong connection between latents, $Z$, and the concepts, $C$. This means that one wants $Z$ to be maximally useful in shaping the concepts $C$, while also ensuring that the concepts are informative about the target.

Moreover, in the CBMs formulation, the concepts come from processing the latent representations, i.e., $c = h(z)$. Thus, due to the data processing inequality, $I(X;C) \le I(X;Z)$, we can bound of the concepts' information bottleneck loss (3)

$$I(Z;C) + I(C;Y) - \beta I(X;C) \ge I(Z;C) + I(C;Y) - \beta I(X;Z). \qquad (4)$$

Thus, our objective is to maximize the upper bound of the concepts' information bottleneck

$$\mathcal{L}_{\text{UB-CIB}} = I(Z;C) + I(C;Y) - \beta I(X;C).[1] \qquad (5)$$

We depict our general framework in Fig. 1. We posit that by compressing the information between the data, $X$, and the concepts, $C$, instead of the latent representations, $Z$, we can control the redundant information of the data within the concepts. Consequently, we can obtain more interpretable concepts instead of first compressing the latents and then obtaining the concepts from them. We hypothesize that this compression also prevents data leakage from the data into the concepts that commonly happens when the concepts are processed through the latents alone. Another interpretation of this process is the compression of the information between the data and the concepts through the marginalized latent representations. Thus, we are obtaining a more robust compression since we compute it through all possible latent representations that lead to that concept.

We propose two implementations of our framework by exploring different ways of solving the mutual information based on a variational approximation of the data distribution. We show our modeling assumptions in Fig. 2.

---

[1]Note that one can obtain the same loss if the optimization problem is constrained over the concepts instead, i.e., $\max I(Z;C) + I(C;Y)$ s.t. $I(X;C) \le I_C$. Nevertheless, we present the relation with the traditional compression for completeness.

## 3.1 BOUNDED CIB

We can consider the upper bound to the concept bottleneck loss (5) in terms of the entropy-based definitions of the mutual information. Then, by using a variational approximation of the data distribution, we bound it by

$$
\mathcal{L}_{\text{UB-CIB}} \leq H(Y) + (1 - \beta)H(C) +
$$
$$
\underset{p(c)}{\mathbb{E}} H\left(p(y \mid c), q(y \mid c)\right) + (1 + \beta) \underset{p(z)}{\mathbb{E}} H\left(p(c \mid z), q(c \mid z)\right), \tag{6}
$$

$$
\mathcal{L}_{\text{UB-CIB}} \leq (1 - \beta)H(C) + \underset{p(c)}{\mathbb{E}} H\left(p(y \mid c), q(y \mid c)\right) + (1 + \beta) \underset{p(z)}{\mathbb{E}} H\left(p(c \mid z), q(c \mid z)\right). \tag{7}
$$

We detail this derivation in Appendix A. We can maximize the concepts' information bottleneck by minimizing the cross entropies of the predictive variables, $y$ and $c$, and their corresponding ground truths and by adjusting the entropy of the concepts. The simplified upper bound of the concept information bottleneck is

$$
\mathcal{L}_{\text{SUB-CIB}} = (1 - \beta)H(C) + \underset{p(c)}{\mathbb{E}} H\left(p(y \mid c), q(y \mid c)\right) + (1 + \beta) \underset{p(z)}{\mathbb{E}} H\left(p(c \mid z), q(c \mid z)\right). \tag{8}
$$

We term the model that uses this bounded concept information bottleneck (8) as $\text{CIBM}_B$. To implement it, we need to estimate the entropy of the concepts distribution $p(c)$. We give details of this estimator in Appendix B.2.

## 3.2 ESTIMATOR-BASED CIB

Another way to obtain a bound over the concept information bottleneck (5) is to only expand the conditional entropies that are not marginalized (A.1) to avoid widening the gap in the bound. That is

$$
\mathcal{L}_{\text{UB-CIB}} = H(Y) + H(C) + \underset{p(c)}{\mathbb{E}} H\left(p(y \mid c), q(y \mid c)\right) + \underset{p(z)}{\mathbb{E}} H\left(p(c \mid z), q(c \mid z)\right) - \beta I(X; C). \tag{9}
$$

If we treat the entropies of the concepts and the labels as constants, we obtain

$$
\mathcal{L}_{\text{E-CIB}} = \underset{p(c)}{\mathbb{E}} H\left(p(y \mid c), q(y \mid c)\right) + \underset{p(z)}{\mathbb{E}} H\left(p(c \mid z), q(c \mid z)\right) + \beta\left(\rho - I(X; C)\right), \tag{10}
$$

where $\rho$ is a constant. We term the model that uses this loss as $\text{CIBM}_E$ since it relies on the estimator of the mutual information. We detail the estimator we used in our implementation in Appendix B.2.

This loss is similar to the one proposed by Kawaguchi et al. (2023), $\mathcal{L}_{\text{K}} = \mathbb{E}_{p(z)} H\left(p(y \mid z), q(y \mid z)\right) + \beta(\rho - I(Z; X))$, if one extends the mutual information from the labels into the concepts in a similar way. In other words, our mutual information estimated loss (10) resembles that of Kawaguchi et al.'s (2023) proposal with the corresponding conditioning changes in the labels and the concepts. Thus, it is interesting to see that other optimization approaches emerge out of this bound. We highlight that our proposal is a generalized framework that encompass a wide range of possible implementations.

Unlike $\mathcal{L}_{\text{SUB-CIB}}$ (8), which simplifies the mutual information terms into cross-entropy losses, $\mathcal{L}_{\text{E-CIB}}$ retains an explicit control over $I(X; C)$. This allows for more granular control over the information flow from inputs to concepts, leading to a tighter constraint on concept leakage. As we show in the results (Table 1), this additional control translates to improved performance in both concept and class prediction accuracy, cf. Section 4.

## 4 EXPERIMENTS

In this experimental section, we aim to evaluate the Concept's Information Bottleneck Models (CIBMs), particularly assessing concept prediction performance and concept leakage in comparison to traditional CBMs using models of similar complexity. Our main goal is to enhance the predictability of concepts, not necessarily to improve target prediction accuracy. Moreover, we investigate the information flows within both CBMs and CIBMs to understand mutual information behavior, alongside testing interventions and applying our proposed metrics to evaluate model efficacy.

We present all implementation details in Appendix B. In the experiments below, for baselines to evaluate our proposal, we compare against CBM (Koh et al., 2020), ProbCBM (Kim et al., 2023),

**Table 1:** Accuracies for our proposed methods, $\text{CIBM}_B$ and $\text{CIBM}_E$, on CUB dataset (avg. 3 runs).

| Method | | Concept | Class |
|---|---|---|---|
| $\text{CIBM}_B$ (vanilla) | | 0.934 | 0.608 |
| | (clip_norm $= 1.0$) | 0.947 | 0.660 |
| | (clip_norm $= 0.1$) | 0.947 | 0.646 |
| | (stop grad. from $H(C)$ into $p(z \mid x)$) | **0.959** | 0.726 |
| $\text{CIBM}_E$ | | **0.959** | **0.729** |

CEM (Zarlenga et al., 2022), and PCBM (Yuksekgonul et al., 2023). We also evaluate a "black-box model" that denotes a model with an architecture equivalent to that of our $\text{CIBM}_E$'s model, without our proposed losses, and trained only to predict class labels, thus, unable to predict concepts that we deemed as a gold-standard for classification.

## 4.1 Datasets

We benchmark our approach on 3 datasets: CUB (Wah et al., 2011), AwA2 (Xian et al., 2019), and aPY (Farhadi et al., 2009). While CUB is a recognized dataset for comparing concept-based approaches (Koh et al., 2020; Kim et al., 2023; Zarlenga et al., 2022), we add the other two datasets for additional evaluations and analysis.

**CUB.** Caltech-UCSD Birds dataset (Wah et al., 2011) is a dataset of birds images totaling in 11788 samples for 200 species. Following Koh et al.'s (2020) work, for reproducibility, we reduce instance-level concept annotations to class-level ones with majority voting. We then keep only the concept that are annotated as present in 10 classes at least after the described voting, resulting in 112 concepts instead of 312. We also employ train/val/test splits provided by Koh et al. (2020), operating with 4796 train images, 1198 val images and 5794 test images. To diversify training data, we augment the images with color jittering and horizontal flip, and resize the images to $299 \times 299$ pixels for the InceptionV3 backbone. Concept groups are obtained by common prefix clustering.

**AwA2.** Animals with attributes dataset (Xian et al., 2019) is a dataset of 37322 images of 50 animal species. For the concepts set, we follow Kim et al.'s (2023) work and keep only the 45 concepts which could be observed on the image. We use ResNet18 embeddings provided by the dataset authors and train FCN on top of them. No additional augmentations are applied to those embeddings.

**aPY.** This is a dataset (Farhadi et al., 2009) of 32 diverse real-world classes we used for proof of concept. We split the dataset into 7362 train, 3068 validation and 4909 test samples stratified on target labels. We train FCN on top of ResNet18 embeddings of input images provided by the dataset authors (Xian et al., 2019). No additional augmentations are applied to those embeddings.

## 4.2 Comparison of different versions of CIB

In Table 1, we compare the performance of $\text{CIBM}_B$ and $\text{CIBM}_E$ on concept and class prediction accuracy for the CUB dataset—using $\beta = 0.5$. As shown, $\text{CIBM}_E$, which retains an explicit mutual information term $I(X; C)$, outperforms $\text{CIBM}_B$ when trained in a fair setup (vanilla) in both metrics. We found that the lack of performance of vanilla $\text{CIBM}_B$ comes from instabilities during training in the latent representations encoder $p(z \mid x)$. We hypothesize that the gradient from the $H(C)$ in the loss (8) damages the feature encoder $p(z \mid x)$ since the entropy is computed w.r.t. the generative concepts $p(c)$ instead of the variational approximated ones $q(c)$. To alleviate this problem, we experimented gradient clipping as well as stopping the gradient from $H(C)$ into the encoder. We found that the latter perform on par with $\text{CIBM}_E$. In the following, we refer to $\text{CIBM}_B$ as the version with stop gradient on it. Overall, $\text{CIBM}_E$'s more granular control over information flow limits concept leakage, results in better accuracies for concepts and labels in comparison to the baselines (cf. Table 2) without changes to its training framework.

These results supports our earlier discussion that the direct estimation of $I(X; C)$ leads to more effective use of concepts in downstream tasks without further changes to the training regime. Nevertheless, with a correctly regularized feature encoder $p(z \mid x)$, a simple estimation in $\text{CIBM}_B$ can achieve similar levels of information gain and accuracy.

### 4.3 Performance across all datasets

We present the evaluation results across three datasets in Table 2. Our "black-box model" serves as a gold standard, representing the highest possible class accuracy achievable by a model similar to ours within a traditional setup that does not provide explanations. We compare against hard (H) and soft (S) CBMs trained jointly (J) or independently (I) (Havasi et al., 2022). In the following, when we refer only to the CBM, we mean the soft joint (SJ) version of it which is closer to our setup. Our main objective is to demonstrate that CIBMs maintain or improve the target prediction accuracy in comparison to CBMs and CEM while improving the concept prediction accuracy and reducing concept leakage. The latter is of particular importance to guarantee the explainability of the results.

Our proposed methods, $\text{CIBM}_B$ and $\text{CIBM}_E$, show an improvement over most methods (apart from CEM) regarding target prediction accuracy for the *CUB dataset*. These improvements come alongside enhanced concept accuracy, thus, realizing the fundamental goal of our approach: to simultaneously boost performance and interpretability. Although we fall short of CEM's class prediction accuracy, our concept prediction accuracy is superior. As for the *AwA2 dataset*, the target accuracy gain is less marked compared to the other datasets but is nevertheless statistically significant. We ascribe this to the dataset's relative simplicity, which narrows the room for enhancement. In the more varied real-world classes of the *aPY dataset*, $\text{CIBM}_E$ significantly outperforms the baseline CBMs in target accuracy. The black-box model may achieve marginally better target accuracy, yet it falls short on interpretability, which is paramount in real-world applications where explanations are necessary.

**Table 2:** Accuracy for CUB, AwA2, and aPY datasets. The results include mean and std. over 5 runs. We report results for different lagrange multipliers $\beta$ for our methods, $\text{CIBM}_B$ and $\text{CIBM}_E$. Black-box is a gold standard for class prediction that offers no explainability over the concepts.

| Data | Method | Concept | Class |
|---|---|---|---|
| CUB | Black-box | – | 0.919±0.002 |
| | CBM (HJ) | 0.956±0.001 | 0.650±0.002 |
| | CBM (HI) | 0.956±0.001 | 0.644±0.001 |
| | CBM (SJ) | 0.956±0.001 | 0.708±0.006 |
| | CEM | 0.954±0.001 | **0.759±0.002** |
| | ProbCBM | 0.956±0.001 | 0.718±0.005 |
| | PCBM | – | 0.610±0.010 |
| | $\text{CIBM}_B$ ($\beta = 0.25$) | 0.958±0.001 | 0.726±0.003 |
| | ($\beta = 0.50$) | 0.958±0.001 | 0.725±0.004 |
| | $\text{CIBM}_E$ ($\beta = 0.25$) | 0.958±0.001 | 0.728±0.005 |
| | ($\beta = 0.50$) | **0.959±0.001** | 0.729±0.003 |
| AwA2 | Black-box | – | 0.893±0.000 |
| | CBM (HJ) | 0.979±0.000 | 0.853±0.002 |
| | CBM (HI) | 0.979±0.000 | 0.836±0.001 |
| | CBM (SJ) | 0.979±0.000 | 0.876±0.001 |
| | CEM | 0.979±0.000 | 0.884±0.002 |
| | PCBM | – | 0.862±0.003 |
| | $\text{CIBM}_B$ ($\beta = 0.25$) | **0.980±0.000** | **0.886±0.002** |
| | ($\beta = 0.50$) | 0.979±0.000 | 0.885±0.002 |
| | $\text{CIBM}_E$ ($\beta = 0.25$) | **0.980±0.000** | 0.885±0.001 |
| | ($\beta = 0.50$) | 0.979±0.000 | 0.883±0.001 |
| aPY | Black-box | – | 0.866±0.003 |
| | CBM (SJ) | **0.967±0.000** | 0.797±0.007 |
| | CEM | **0.967±0.000** | **0.870±0.003** |
| | $\text{CIBM}_B$ ($\beta = 0.25$) | **0.967±0.000** | 0.850±0.006 |
| | ($\beta = 0.50$) | **0.967±0.000** | 0.856±0.005 |
| | $\text{CIBM}_E$ ($\beta = 0.25$) | **0.967±0.000** | 0.858±0.004 |
| | ($\beta = 0.50$) | **0.967±0.000** | 0.856±0.004 |

The rise in concept accuracy relative to existing methods highlights the advantages of our mutual information regularization. This approach helps stop concept leakage and ensures that concepts are both informative and closely tied to the final prediction. This finding is consistent with our theoretical framework, which advocates that controlling the information flow between inputs and concepts through the Information Bottleneck can yield more interpretable and significantly meaningful concepts without compromising performance. Importantly, our approach maintains concept accuracy, suggesting that the mutual information regularization effectively curtails concept leakage even in less complex tasks. This is consistent with our theoretical model, which maintains that minimizing $I(X;C)$ ensures that only pertinent information is channeled through the concepts, thus, increasing the robustness across various datasets.

### 4.4 Information Flow in CIB

We analyze the flow of information between inputs, $X$, latents, $Z$, concepts, $C$, and labels, $Y$, and present them in Fig. 3 for the CUB and AwA2 datasets. The objective of the information plane is to show the mutual information on the model variables after training. In particular, we expect to see a model with high $I(Z;C)$ and $I(C;Y)$ such that the corresponding variables are dependent on each other (maximally expressive), and simultaneously, low $I(X;C)$ and $I(X;Z)$ to show that the corresponding variables are maximally compressive. However, the compression of the variables

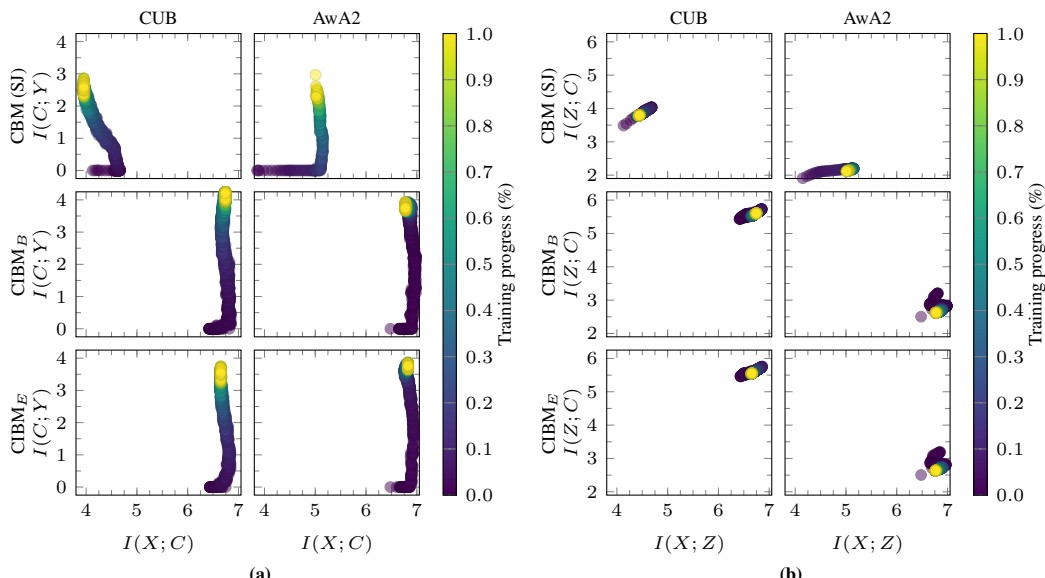

**Figure 3:** Memorization-compression pattern in the information flow (in nats) of original CBM (SJ) and our proposed methods, $CIBM_B$ and $CIBM_E$. Warmer colors denote later steps in training. We show the information plane of (a) the variables $X$, $C$, and $Y$; and (b) the variables $X$, $Z$, and $C$.

alone, minimal $I(X;C)$ or $I(X;Z)$, does not guarantee that the important parts of the variables are being compressed and retained.

CIBMs achieve higher mutual information $I(C;Y)$ while aligning with the target labels—as shown in the prediction tasks in Table 2. In contrast to CBMs, which exhibit lower mutual information between inputs and latent and concept representations, $I(X;Z)$ and $I(X;C)$, CIBMs mutual information w.r.t. the inputs $X$ is higher than the CBMs. This behavior reflects the fact that CIBMs are optimized to retain task-relevant information while removing irrelevant or redundant information but not necessarily compressing as much—reflected in the higher $I(X;Z)$ and $I(X;C)$. Thus, lower mutual information $I(X;Z)$ and $I(X;C)$ in CBMs does not necessarily indicate better compression given its lower predictive accuracy. Instead, it may reflect a failure to capture meaningful input features, resulting in noisier or less predictive concepts. Moreover, we note that the plots in Fig. 3(b) for $CIBM_B$ and $CIBM_E$ look similar but they differ in hundredths.

To demonstrate the effects of the compression patterns, we evaluate the alignment between representations and the target $I(C;Y)$ and show that CIBMs consistently outperform CBMs, indicating that the retained information is both relevant and predictive—cf. Section 4.3. Additionally, CIBMs achieve better interpretability and concept quality, reinforcing that the higher mutual information is a reflection of meaningful expressiveness rather than leakage—cf. Section 4.5. This is further supported by the proposed intervention-based metrics ($AUC_{TTI}$ and $NAUC_{TTI}$) which highlight the importance of retaining task-relevant information in the concepts $C$. While CBMs exhibit lower mutual information between inputs and representations, $I(X;C)$ and $I(X;Z)$, their poorer performance on these metrics, particularly under concept corruption, suggests that this lower information content stems from a failure to capture sufficient relevant features. By contrast, the higher $I(X;C)$ and $I(X;Z)$ in our CIBMs reflect the retention of meaningful pieces that contribute to better concept quality and downstream task performance. These findings demonstrate that reducing concept leakage requires selectively preserving relevant information rather than minimizing mutual information indiscriminately.

## 4.5 INTERVENTIONS

A key advantage of CBMs is their ability to perform *test-time interventions*, allowing users to correct predicted concepts and improve the models final decisions. To demonstrate that our model effectively utilizes concept information and avoids concept leakage, we simulate interventions by replacing predicted concepts with their ground truth values. Following prior work, we intervene on

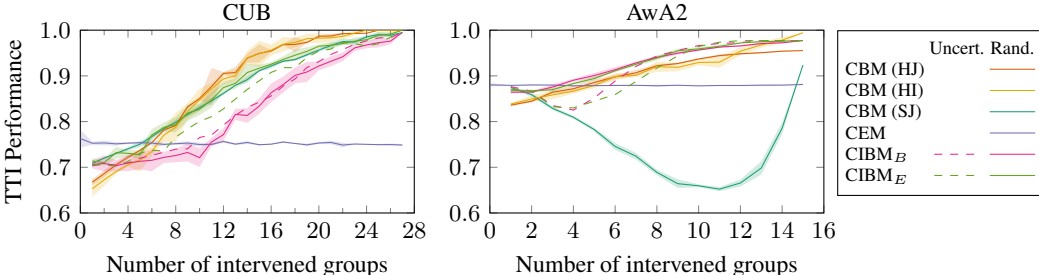

**Figure 4:** Change in target prediction accuracy after intervening on concept groups following the strategies "uncertainty" and "random" as described in Section 4.5. (TTI stands for Test-Time Interventions.)

*groups of concepts* rather than individual concepts, leveraging this strategy to assess how cumulative corrections impact target performance (Koh et al., 2020; Kim et al., 2023). We, then, plot the target task performance improvement against number of concept groups intervened. The resulting curve is denoted as the interventions curve.

We implement two ways of choosing a set of concept groups to intervene on: (i) Random: Concept groups are randomly selected for intervention, and results are averaged over five runs to account for variability. (ii) Uncertainty Based: Since our model predicts parameters of a Gaussian distribution for concept logits, we can measure the likelihood of zero for each concept in that distribution. Zero acts as a threshold for prediction and indicates complete uncertainty about the presence or absence of a concept. We calculate the mean of these likelihoods for a group of concepts and define it as the groups uncertainty. We then select the groups with the highest uncertainty for intervention. We highlight that this setup is not directly applicable to CBMs.

Figure 4 shows that as more concept groups are intervened upon, in CUB, performance improves consistently, demonstrating that the model *relies on accurate concept information* and does not suffer from concept leakage. The steady improvement confirms the model's ability to be "debugged" by correcting concept predictions. In AwA2, our methods have a small dip observed for uncertainty-based selection is likely due to imperfect uncertainty estimation, but overall, interventions still significantly boost performance. On the contrary, soft CBMs shows a higher dip in performance which may be due to the random strategy for the interventions.

While hard CBMs can excel during large-scale interventions due to binary concept representations having low information leakage, their performance is initially lower than CIBMs due to rigid concept representations, making them less adaptive to partial interventions and coarser datasets like AwA. Moreover, CIBMs consistently deliver stable intervention performance across datasets and achieve superior target prediction accuracy compared to all CBM variants, including hard, joint, and independent training configurations. We provide a full analysis in Appendix D. Despite the performance in these interventions, the hard CBMs performance suffers and they are outperformed by our CIBMs—cf. Table 2.

### 4.6 CONCEPT SET GOODNESS MEASURE

In CBMs, the quality of the concept set is crucial for accurate downstream task predictions. However, there is a lack of effective metrics to reliably assess concept set goodness. Existing metrics, such as the Concept Alignment Score, proposed by Zarlenga et al. (2022), evaluate whether the model has captured meaningful concept representations but do not explicitly measure how well these concepts improve downstream task performance during interventions. Moreover, they are tuned for CEM and do not extend beyond it.

We address this gap by proposing two metrics: as area under interventions curve, and the area under curve of relative improvements. Denote by $\mathcal{I}(x)$ the model's performance for $x$ concept groups used in the intervention. Then the Test-Time Interventions accuracy is

$$\text{AUC}_{\text{TTI}} = \frac{1}{n} \sum_{i=1}^{n} \mathcal{I}(i), \tag{11}$$

Table 3: Change in interventions performance with concept set corruption.

| | CUB | | | | | |
|---|---|---|---|---|---|---|
| | AUC | | | NAUC | | |
| Corrupt | CBM | $\text{CIBM}_B$ | $\text{CIBM}_E$ | CBM | $\text{CIBM}_B$ | $\text{CIBM}_E$ |
| 0 | 54.374 | 65.644 | 64.634 | 0.001260 | 0.001481 | 0.001432 |
| 4 | 53.135 | 64.519 | 63.464 | 0.001198 | 0.001525 | 0.001487 |
| 8 | 51.291 | 53.135 | 60.202 | 0.001166 | 0.001198 | 0.001444 |
| 16 | 50.694 | 60.240 | 59.424 | 0.001068 | 0.001388 | 0.001349 |
| 32 | 46.101 | 52.956 | 51.258 | 0.000863 | 0.001298 | 0.001231 |
| 64 | 32.069 | 30.582 | 29.271 | -0.000339 | 0.000571 | 0.000504 |

| | AwA2 | | | | | |
|---|---|---|---|---|---|---|
| | AUC | | | NAUC | | |
| Corrupt | CBM | $\text{CIBM}_B$ | $\text{CIBM}_E$ | CBM | $\text{CIBM}_B$ | $\text{CIBM}_E$ |
| No | 84.753 | 91.573 | 92.225 | 0.002808 | 0.005350 | 0.006250 |
| Yes | 83.985 | 90.631 | 90.879 | 0.004484 | 0.005218 | 0.006474 |

and the normalized version of the Test-Time Interventions accuracy is

$$\text{NAUC}_{\text{TTI}} = \frac{1}{n} \sum_{i=1}^{n} \left( \mathcal{I}(i) - \mathcal{I}(i-1) \right). \tag{12}$$

The idea behind these measures is simple: if a concept set is of high quality, the task accuracy will steadily approach $100\%$ as more concept groups are intervened upon, resulting in a large area under the curve. Conversely, if the concept set is incomplete or noisy, performance gains will be limited, even with multiple interventions, which can indicate concept leakage.

The latter expression (12) could be simplified to just scaled difference between a model with full concept set used for interventions and performance of a model with no interventions, however, the meaning it has is how much does the performance change per one group added to the interventions pool. To test this, we generate corrupted concept sets by replacing selected concepts with noisy ones. Importantly, we maintain the original groupings of concepts.

Table 3 shows the results of our metrics, and we show the disaggregated plots in Fig. C.1. The number in the "corrupt" column denotes the number of concepts replaced with random ones for CUB, and for AwA2 "No" denotes a clear concept set and "Yes" denotes a concept set with one concept changed to corrupt. As expected, performance drops with corrupt concepts, since they contain no useful information for the target task. One consequence of our training is that if one has two concept annotations for some dataset, then it is possible to use CIBMs performance to determine which concept set is better.

Our results demonstrate that $\text{CIBM}_E$ is more sensitive to concept quality compared to vanilla CBM, making it a better indicator of concept set reliability. Negative values in normalized intervention AUC indicate possible concept leakage.

## 5 CONCLUSION

In this paper, we integrated the IB with CBMs and proposed a first-principled theoretical framework to understand CIBMs, resulting in enhanced concept performance, reduced concept leakage, and maintained accuracy in target predictions compared to similar models. We developed two model variants that have complementary performances dependent on the estimators used. Our methods were validated on popular CBM datasets. We proposed new metrics to accurately evaluate concept set quality and examined information flow within our IB-enhanced CBM objectives. Our findings suggest that conventional CBMs might compress useless information, whereas our regularization approach achieves better predictive accuracy with less compression. Furthermore, we assessed our model's interpretability and capacity for intervention, showing that our IB objective retains or even enhances performance when interventions are applied.

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

# A   DETAILED DERIVATION OF CIB

In this section we present the detailed derivations to obtained the results described in Section 3.1.

We can re-write the upper bound of the concepts' information bottleneck as

$$\mathcal{L}_{\text{UB-CIB}} = H(Y) + (1 - \beta)H(C) - H(Y \mid C) - H(C \mid Z) - \beta H(C \mid X) \tag{A.1}$$

to work with the entropies instead. To find a more suitable form to tackle this bound, we consider an approximation of the predictors for the labels and the concepts, $q(y \mid c)$ and $q(c \mid z)$, based on two variational distributions that will be implemented through neural networks—cf. Fig. 2. Consider, on one hand,

$$H(Y \mid C) = \iint dy \, dc \, p(y, c) \log p(y \mid c), \tag{A.2a}$$

$$= \iint dy \, dc \, p(y, c) \log \left[ p(y \mid c) \frac{q(y \mid c)}{q(y \mid c)} \right], \tag{A.2b}$$

$$= \iint dy \, dc \, p(y \mid c)p(c) \left[ \log \frac{p(y \mid c)}{q(y \mid c)} + \log q(y \mid c) \right], \tag{A.2c}$$

$$= \int dc \, p(c) \int dy \, p(y \mid c) \left[ \log \frac{p(y \mid c)}{q(y \mid c)} + \log q(y \mid c) \right], \tag{A.2d}$$

$$= \mathbb{E}_{p(c)} \left[ \text{KL}\big(p(y \mid c) \,\big\|\, q(y \mid c)\big) - H\left(p(y \mid c), q(y \mid c)\right) \right]. \tag{A.2e}$$

We introduce the variational distribution $q(y \mid c)$ to obtain the cross-entropy w.r.t. the ground truth and this results on an additional term to make the variational distribution close to the prior. In other words, we can interpret the conditional entropy of the labels w.r.t. the concepts as an optimization of the variational distribution $q(y \mid c)$ with the true conditional of the labels given the concepts $p(y \mid c)$ through a Kullback-Leibler divergence (KL) and the cross-entropy between them. This last cross-entropy can be interpreted as the traditional prediction loss of the true labels and the predicted ones. Similarly,

$$H(C \mid Z) = \iint dc \, dz \, p(c, z) \log p(c \mid z), \tag{A.3a}$$

$$= \iint dc \, dz \, p(c, z) \log \left[ p(c \mid z) \frac{q(c \mid z)}{q(c \mid z)} \right], \tag{A.3b}$$

$$= \iint dc \, dz \, p(c \mid z)p(z) \left[ \log \frac{p(c \mid z)}{q(c \mid z)} + \log q(c \mid z) \right], \tag{A.3c}$$

$$= \int dz \, p(z) \int dc \, p(c \mid z) \left[ \log \frac{p(c \mid z)}{q(c \mid z)} + \log q(c \mid z) \right], \tag{A.3d}$$

$$= \mathbb{E}_{p(z)} \left[ \text{KL}\big(p(c \mid z) \,\big\|\, q(c \mid z)\big) - H\left(p(c \mid z), q(c \mid z)\right) \right], \tag{A.3e}$$

were $q(c \mid z)$ is a variational distribution that predicts the concepts given the latent representations. This decomposition of the conditional entropy of the concepts given the representations follows the same principles as the conditional of the labels given the concepts (A.2). On the other hand, the conditional entropy of the concepts w.r.t. the data is bounded due to the marginalization of the latent representations on their dependency. That is,

$$H(C \mid X) = \iint dc\,dx\,p(c,x) \log p(c \mid x), \tag{A.4a}$$

$$= \iint dc\,dx\,p(c,x) \log \int dz\,p(c,z \mid x), \tag{A.4b}$$

$$= \iint dc\,dx\,p(c,x) \log \int dz\,p(c \mid z)p(z \mid x), \tag{A.4c}$$

$$\leq \iint dc\,dx\,p(c,x) \int dz\,p(z \mid x) \log p(c \mid z), \tag{A.4d}$$

$$= \iiint dc\,dz\,dx\,p(c,z,x) \int dz\,p(z \mid x) \log p(c \mid z), \tag{A.4e}$$

$$= \iiint dc\,dz\,dx\,p(c \mid z)p(z \mid x)p(x) \int dz\,p(z \mid x) \log p(c \mid z), \tag{A.4f}$$

$$= \int dx\,p(x) \iiint dc\,dz^2\,p(c \mid z)p(z \mid x)^2 \log p(c \mid z), \tag{A.4g}$$

$$= \int dx\,p(x) \iiint dc\,dz^2\,p(c \mid z)p(z \mid x)^2 \log \left[ p(c \mid z) \frac{q(c \mid z)}{q(c \mid z)} \right], \tag{A.4h}$$

$$= \int dx\,p(x) \iint dz^2\,p(z \mid x)^2 \int dc\,p(c \mid z) \log \left[ p(c \mid z) \frac{q(c \mid z)}{q(c \mid z)} \right], \tag{A.4i}$$

$$= \mathop{\mathbb{E}}_{p(x)} \mathop{\mathbb{E}}_{p(z \mid x)} \int dc\,p(c \mid z) \log \left[ p(c \mid z) \frac{q(c \mid z)}{q(c \mid z)} \right], \tag{A.4j}$$

$$= \mathop{\mathbb{E}}_{p(z \mid x)p(x)} \int dc\,p(c \mid z) \left[ \log \frac{p(c \mid z)}{q(c \mid z)} + \log q(c \mid z) \right], \tag{A.4k}$$

$$= \mathop{\mathbb{E}}_{p(z \mid x)p(x)} \left[ \mathrm{KL}\big(p(c \mid z) \,\|\, q(c \mid z)\big) - H\left(p(c \mid z), q(c \mid z)\right) \right], \tag{A.4l}$$

where the bound comes from applying the Jensen's inequality. Thus, the upper bound to the concept bottleneck loss (5), given that we remove the KLs constraints, due to their positivity, from the conditional entropies (A.2), (A.3) and (A.4) is

$$\mathcal{L}_{\text{UB-CIB}} \leq H(Y) + (1-\beta)H(C) + \mathop{\mathbb{E}}_{p(c)} H\left(p(y \mid c), q(y \mid c)\right) + (1+\beta) \mathop{\mathbb{E}}_{p(z)} H\left(p(c \mid z), q(c \mid z)\right). \tag{A.5}$$

The bound gap can be further reduced by dropping the entropy of the labels as

$$\mathcal{L}_{\text{UB-CIB}} \leq (1-\beta)H(C) + \mathop{\mathbb{E}}_{p(c)} H\left(p(y \mid c), q(y \mid c)\right) + (1+\beta) \mathop{\mathbb{E}}_{p(z)} H\left(p(c \mid z), q(c \mid z)\right), \tag{A.6}$$

$$= \mathcal{L}_{\text{SUB-CIB}}. \tag{A.7}$$

In other words, we can maximize the concepts' information bottleneck by minimizing the cross entropies of the predictive variables, $y$ and $c$, and their corresponding ground truths and by adjusting the entropy of the concepts.

## B  IMPLEMENTATION DETAILS

### B.1  DETAILS ON THE MODELS

For CUB dataset, we choose InceptionV3 as image embedder ($p(z \mid x)$). We add on top of its embeddings two 1-layer MLP (for mean and std in the variational approximation $q(c \mid z)$) each of dimensionality 112—the number of concepts left after filtration identical to one done in Koh et al. (2020). We obtain concept logits as $C = \text{pred}_\mu(x) + \text{pred}_\sigma(x) \cdot \epsilon$, where $\epsilon$ is a random standard Gaussian noise. On top of concepts logits, we stack label predictor $q(y \mid c)$ (also 1-layer MLP).

```python
class CIBM:
    def __init__(num_concepts=112, num_labels=200):
        backbone = inceptionv3()
        pred_mu = Linear(2048, num_concepts)  # 2048 is the embedding dim
            ↪ of inceptionv3
        pred_sigma = Sequential(Linear(2048, num_concepts), Softplus)
        pred_label = Linear(num_concepts, num_labels)

    def forward(x):
        z = backbone(x)
        z = ReLU(z)
        eps = N(0; I).sample(num_samples=len(x))
        mu = pred_mu(z)
        sigma = pred_sigma(z).clamp(min=1e-7)  # for numeric stability
        c = mu + sigma * eps
        y = pred_label(c)
        return c, y
```

**Listing B.1:** CIBM Python code.

All activations between the layers are ReLU. The overall code would like the example shown in Listing B.1.

For AwA2 and aPY the only difference is that we use on pre-computed embeddings from ResNet18 without training the backbone.

For CEM (Zarlenga et al., 2022) there are basically two training options: intervention-aware and basic. In the latter, the model just optimizes two CE objectives. We implemented and trained the basic setup on CUB, AwA2, and aPY. Then, we measured the interventions performance.

Our accuracies coincided with those reported by Zarlenga et al. (2022) in their paper on CUB dataset. And intervention performance of this intervention-unaware model variant matched the reported behavior from the authors (i.e., no gain from interventions).

## B.2 ESTIMATORS DETAILS

**Mutual Information Estimator.** Before each gradient update, we compute cross-entropies over the current batch $B_c$, and then randomly sample batch $B_c'$ from the training dataset to estimate $I(X; C)$ on this batch.

Our mutual information estimator is taken from Kawaguchi et al. (2023). We rely on the fact that concepts logits have Gaussian distribution for estimation of $\log p(c \mid x)$. And then, we use the random samples $B_c'$ to approximate the marginal of the concepts $\log p(c)$. The mutual information $I(C; X)$ is then a Monte-Carlo estimate of $\log p(c \mid x) - \log p(c)$.

**Entropy Estimator.** Since concepts $C$ are distributed normally, we use $H(C) = \frac{D}{2}(1 + \log(2\pi)) + \frac{1}{2}\log|\Sigma|$. For simplicity (since the number of concepts $D$ is constant throughout the training and inference) we use $\hat{H}(C) = \frac{1}{2}\log|\Sigma| = \sum \log(\sigma_i)$ since $\Sigma$ is a diagonal matrix in our setup.

## B.3 TRAINING PARAMETERS

We set batch size to 128 and number of samples for MI estimation to 64. For all experiments we used Adam (Kingma & Ba, 2015) optimizer with $lr = 0.003$ and $wd = 0.001$. We experimented with gradient clipping, but it led to either slow or divergent training, so we are not clipping the gradients in any of the experiments.

## B.4 DETAILS ON EXPERIMENTS

The image embedder backbone is only trained for CUB dataset, and for AwA2 and aPY we use pre-computed image embeddings. The ground truth concept labels are binary across all dataset, but

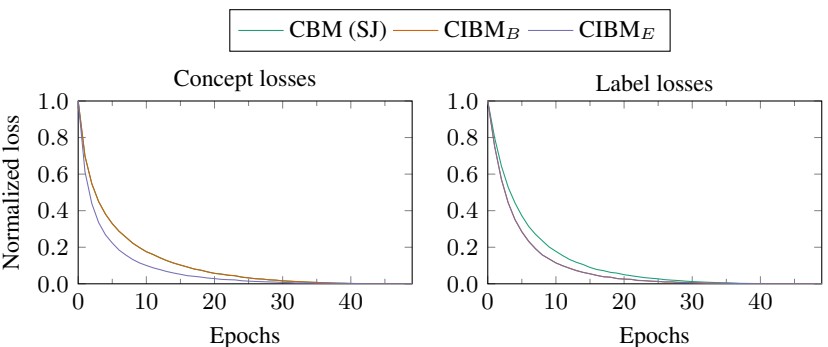

**Figure B.1:** Losses on the validation set of CUB for different methods.

concepts predictions passed to label classifier are non-binary: we are training only (and comparing only against) models using soft concepts for class prediction.

When training $\text{CIBM}_B$, we used the $\mathcal{L}_{\text{SUB-CIB}}$ (8) for better performance. We backpropagate the gradients from the cross-entropies over concepts and labels through the entire network—both backbone $q(c \mid z)$ and MLPs on top of the encoder $q(y \mid c)$. For $H(C)$, however, the situation is different: gradients from this part of the loss function are propagated only through the MLPs, $q(c \mid z)$ and $q(y \mid c)$, but not the image embedder backbone $p(z \mid x)$. We found that such (partial) "freezing" of the encoder with respect to $H(C)$ constraint dramatically improves the quality of both concepts and labels prediction. While we do not have access to the ground truth probability distribution for the concepts $p(c \mid z)$, we have access to the ground truth concept labels. Our implementation uses the a supervised cross-entropy using the ground truth labels. The concepts' predictor can be seens as a multi-label task classifier. In practice, we compute $C$ logits, then, we compute binary cross-entropy (BCE) for each of these logits with binary labels. Finally, we backpropagate them through the means of BCEs.

We show the normalized loss function values on the validation set of CUB in Fig. B.1 to show the convergence of CIBMs in comparison to CBM. Note that visually the concept losses on between CBM and $\text{CIBM}_E$ and the label losses between CIBMs are similar, but they differ slightly.

## C  ADDITIONAL RESULTS

In Fig. C.1, we show additional results about the aggregated interventions that we dicussed in Section 4.6 and that we showed in Table 3. We plot the interventions in the traditional way by showing the intervened groups and the TTI performance for six different corruption settings.

## D  DISCUSSION ABOUT CBMS SETUPS

Hard CBMs use hard concept representations, meaning that instead of producing a probabilistic output (as in soft concepts in soft CBM), each concept prediction is treated as a discrete binary or categorical value. These hard predictions are used as inputs to the downstream task (class prediction), making the pipeline interpretable and less expressive, thus less prone to information leakage.

When compared with soft CBMs and Soft CIBMs:

- Representation:
    - Hard CBMs: Use discrete hard values for concepts (e.g., 0 or 1 for binary concepts).
    - Soft CBMs: Use continuous values (e.g., logits or probabilities).
    - Soft CIBMs: Similar to soft CBMs but use IB to minimize irrelevant information, reducing concept leakage.
- Information Flow:
    - Hard CBMs: Compress information into discrete concept values, which prevents information leakage but risks losing useful details for downstream tasks.

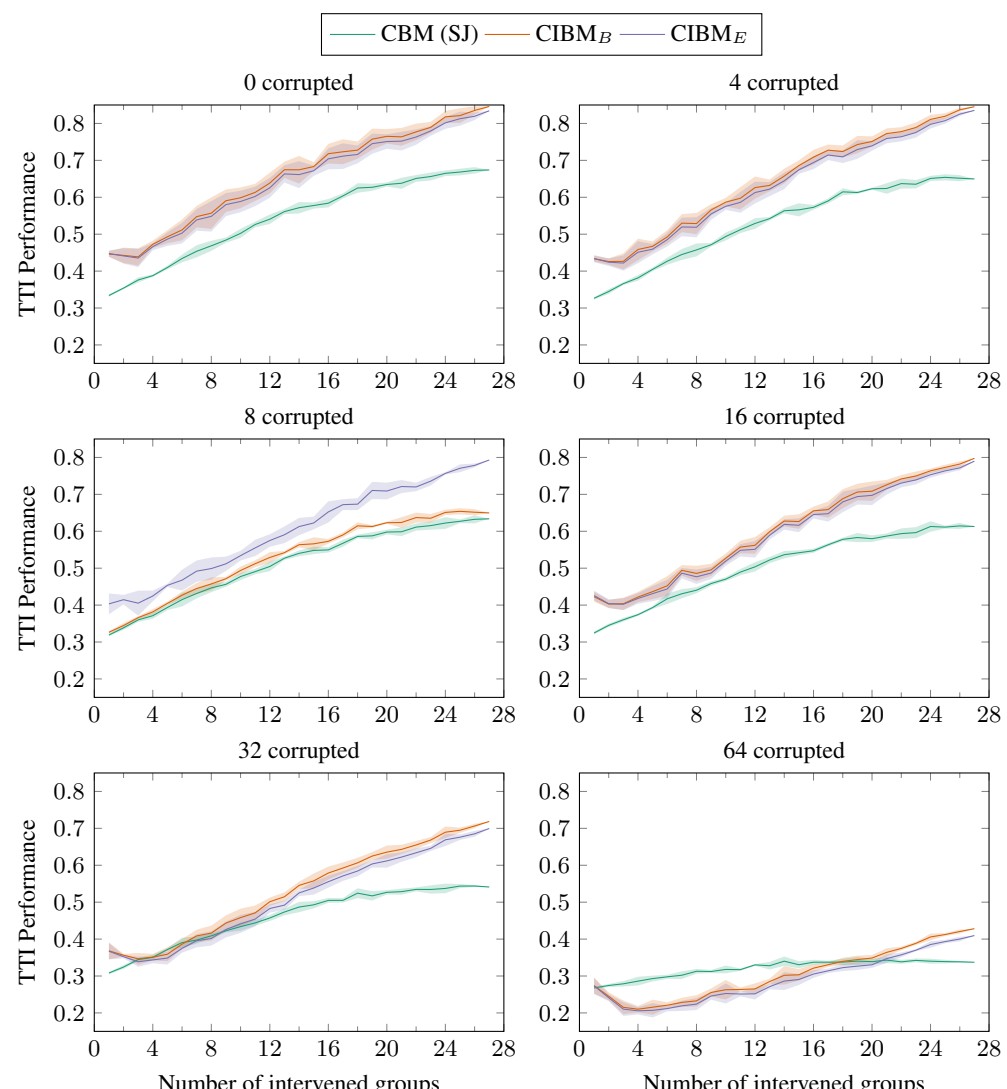

**Figure C.1:** Change in target prediction accuracy for different number of corrupted concepts. These are the expanded results of Table 3. (TTI stands for Test-Time Interventions.)

  – Soft CBMs: Retain richer information but are more prone to concept leakage.

  – Soft CIBMs: Balance retaining relevant information while mitigating leakage through the IB framework.

- Interventions:

  – Hard CBMs: Explicitly rely on discrete corrections during interventions, which can have a significant impact.

  – Soft CBMs and CIBMs: Treat interventions as updates to probabilities or logits, which is more expressive, but could induce noise in concepts.

Due to their rigidity, without enough interventions, hard CBMs cannot recover from errors or noise in the predicted concepts because the discrete pipeline does not allow for soft adjustments.

But, as more concepts are corrected, the discrete nature of hard CBMs becomes an advantage together with its independent training: ground truth, hard values fully override noisy predictions, ensuring perfect input for the downstream classifier, which was previously trained also on ground truth concepts from train set.

Soft CBMs and CIBMs, while retaining more information, still rely on probabilistic updates during interventions, which may not fully override noisy concept predictions.

Overall, CIBMs are superior because they combine the advantages of soft representations (expressiveness, better performance) with mechanisms to mitigate concept leakage (robustness, interpretability). Hard CBMs, while conceptually cleaner in avoiding leakage, fail to achieve the same level of downstream performance and adaptability, particularly in more realistic or challenging scenarios.

