# OpenReview forum: "Concepts' Information Bottleneck Models"
_ICLR.cc/2025/Conference — Submitted to ICLR 2025_

### Official Review · Reviewer_xX2U · 2024-10-24

**Soundness:** 3
**Presentation:** 3
**Contribution:** 3
**Rating:** 6
**Confidence:** 4

**Summary:**

This work proposes an enhancement to Concept Bottleneck Models (CBMs) by integrating the Information Bottleneck (IB) framework, aimed at addressing issues of concept leakage and reduced performance. Further, a model-based metric is introduced to measure concept set goodness.  Experiments conducted on CUB, AwA2, and aPY datasets demonstrate that IB-augmented CBMs improve both concept and target prediction accuracy while increasing intervenability.

**Strengths:**

-	The paper introduces a novel integration of the Information Bottleneck framework into CBMs, which is an interesting theoretical contribution to the area of explainable AI.

-	The paper provides sufficient experimental results on multiple datasets, demonstrating the performance of the proposed method in both concept and target prediction accuracy being on par or slightly better than current approaches

-	The introduction of a novel metric to assess the quality of concept sets based on intervention performance is a valuable addition. This metric offers a direct and interpretable evaluation of concept set goodness, addressing a gap in the current literature.

**Weaknesses:**

- The integration of the Information Bottleneck framework adds complexity to the CBMs. A more detailed discussion of the computational overhead and implementation challenges associated with the proposed method would improve the paper.
-	The performance of the proposed method may be sensitive to the choice of hyperparameters, such as the Lagrangian multiplier β. A more systematic approach to hyperparameter tuning could be explored to optimize performance.

**Questions:**

you claim to achieve significant improvement in performance compared to vanilla CBMs and related advanced architectures. How do you support this claim of being significant? Is this meant to be statistically significant?

---

> ### Author Response · Authors · 2024-11-21
> **Reply to raised comments**
>
> > The integration of the Information Bottleneck framework adds complexity to the CBMs. A more detailed discussion of the computational overhead and implementation challenges associated with the proposed method would improve the paper.
>
> The inclusion of the IB mostly adds training overhead due to the need to compute the additional losses, at inference time there are almost none (only for the variational approximation computation like in a VAE).
>
> In big-O complexity the overhead is not present (as it is a constant multiplier, since for the MI estimator, on each training forward step,  we just need to compute the MI over a random subset of training samples. We set the size of this random subset to batch_size of 2 * batch_size. So where a basic CBM does forward+backward in O(batch_size) operations, we do so in O(2 * batch_size) operations.
>
> ---
> > The performance of the proposed method may be sensitive to the choice of hyperparameters, such as the Lagrangian multiplier β. A more systematic approach to hyperparameter tuning could be explored to optimize performance.
>
> We show the changes in the proposed CIBMs with different $\beta$ parameters (0.25 and 0.50) in the updated manuscript in Table 2.
>
> ---
> > you claim to achieve significant improvement in performance compared to vanilla CBMs and related advanced architectures. How do you support this claim of being significant? Is this meant to be statistically significant?
>
> Yes.  As requested by the reviewer, we performed a Studetn’s paired t-test between the CBM and $\text{CIBM}_E$, and obtained that they are different with a significance p=0.022.

---

> ### Author Response · Authors · 2024-11-26
> **Were your questions addressed?**
>
> Dear reviewer xX2U,
>
> We were wondering if our reply addressed your concerns.
>
> We will appreciate to hear from you since the time to make updates to the paper is running out.

---

> > ### Comment · Reviewer_xX2U · 2024-11-26
> > **Thanks for the rebuttal**
> >
> > I appreciate the rebuttal. The authors have addressed the concerns I raised in my review. However, reading through the other reviews, I think that comparing to hard concept representations would greatly improve the paper, demonstrating the quantitative improvement in concept leakage reduction achieved by the proposed method compared to hard concept representations.

---

> > > ### Author Response · Authors · 2024-11-26
> > > **Follow up about the hard concepts**
> > >
> > > We thank the reviewers for the positive evaluation.  We highlight that we added new results during the rebuttal.  We have added our implementation of CEM results and interventions, expanded corrupted concepts, and now have hard CBMs.

---

> > > > ### Author Response · Authors · 2024-11-27
> > > > **New experiments for hard CBMs**
> > > >
> > > > We were able to get the results of the hard CBMs and put them into the manuscript (Table 2 and Fig. 4).  We also added a discussion about the different training setups and differences with our proposal in Appendix D.
> > > >
> > > > We hope that these comparisons improved our paper as suggested by the reviewer.

---

### Official Review · Reviewer_miqn · 2024-10-24

**Soundness:** 2
**Presentation:** 3
**Contribution:** 3
**Rating:** 6
**Confidence:** 3

**Summary:**

The paper addresses a significant issue in Concept Bottleneck Models (CBMs): concept leakage. This occurs when the model encodes additional information in the concept values beyond what is necessary to solve a task. To mitigate this, the authors propose Concept Information Bottleneck Models (CIBMs), a novel training approach for CBMs that utilizes Information Bottleneck and Mutual Information techniques. By minimizing the information bottleneck between concepts, inputs, and outputs, they effectively limit the information flow, thereby reducing leakage. The framing of this approach is intriguing, and the experimental results provide promising insights into its effectiveness. Additionally, the paper introduces a new metric and its variation for evaluating how well a CBM handles interventions, which is closely related to measuring concept leakage.

**Strengths:**

- The paper is well-written and easy to follow. It provides a solid motivation for the problem, offering sufficient context on concept leakage and how it has been addressed by existing methods.
- Employing Mutual Information is a novel and intriguing approach to mitigate concept leakage, a critical issue in Concept Bottleneck Models (CBMs).
- The authors effectively guide the reader through the solution’s formulation, offering enough theoretical insights to understand why they arrived at the two proposed solutions: $CIBM_E$ and $\text{CIBM}_{\text{B}}$.
- The newly introduced metric is a clever addition, as it provides an automatic evaluation of what prior works have mostly assessed graphically. While the concept itself is not entirely new (as CBMs are often evaluated through plots showing model performance with increasing interventions), the metric encapsulates this idea into a single value that assesses the overall trend.

**Weaknesses:**

- In the practical scenario, the architecture they employ is not entirely clear to me. I understand that the $q(\cdot)$ functions are distributions parameterized by neural networks, but the details regarding the rest of the model, particularly $z$, are unclear (see Q1). Are the concepts being supervised, and is the same set of concepts used as in traditional CBMs? A simple visual representation of the model, highlighting the differences introduced compared to CBMs, would be very helpful.
- The experimental section also raises some concerns:
	1.	The rationale behind dropping certain baselines (as seen in Table 2) is not well explained. For instance, I would have expected to see all baselines, particularly CEM, as it is one of the most powerful CBM-like models in terms of accuracy (see Q2).
	2.	Several claims are either missing supporting information (Figure 1), lack proper motivation (L426-431), or are somewhat misleading (L467-469). Regarding Figure 1, there is no discussion about $I(X;C)$, which, as far as I understood, should exhibit a lower value for CIBM later in the training compared to CBM, but this doesn’t seem to happen and isn’t discussed. Both CBM and CIBM display a similar trend in $I(C;Y)$, though the effect is less pronounced for CBM (as expected) (see Q3). Additionally, the explanation in L426-431 is unclear, especially since Figure 3 shows CBM and CIBM behaving similarly, leaving it unclear what insight the reader is supposed to take away (see Q4). Lastly, L467-469 are somewhat misleading, as there is no baseline comparison. Even a comparison with CBM would be fine here. Since CBM might also exhibit a similar trend in responsiveness to interventions while suffering from leakage, the statement “does not suffer from concept leakage” seems too strong or well not motivated (see Q5).
	3.	If the goal of the model is to reduce leakage, why isn’t it compared against other models that tackle the same issue, such as those cited in the paper (e.g., “Addressing Leakage in Concept Bottleneck Models”)? Including a comparison with at least one of these models would strengthen the experimental validation (see Q6).

Addressing these issues would significantly improve the clarity and strength of the paper, and I would be inclined to raise my score.

**Questions:**

**Q1**: Is $z$ simply a hidden representation extracted from a neural network (e.g., the output of a ResNet)? Does your model follow the structure: $x \rightarrow z \rightarrow c \rightarrow y$? Clarifying this would help improve understanding of the overall architecture.

**Q2**: Why did you drop certain baselines in some experiments, retaining only a few (e.g., dropping CEM in all experiments except CUB)? I would prefer a comparison with the strongest model, such as CEM, instead of weaker models like PCBM, to ensure a fair performance evaluation.

**Q3**: Could you clarify whether the trend or the higher value of $I(C;Y)$ is more significant, and explain why this matters? Additionally, what does a lower $I(X;C)$ represent in practical terms? Moreover, please standardize the x-axis range across all plots to avoid misleading comparisons between methods.

**Q4**: The plots in Figure 3 all appear quite similar, and it’s unclear what specific differences I should focus on. Could you explain your claims more clearly and point out the key takeaways?

**Q5**: Why was CBM not included as a baseline in Figure 4? Given that CBM likely exhibits a similar trend to CIBM, the statement that “CIBM does not suffer from concept leakage” feels unsupported. Could you strengthen this claim with further evidence or comparative results?

**Q6**: Why did you choose not to compare your model with other approaches specifically designed to reduce leakage, such as “Addressing Leakage in Concept Bottleneck Models”?

**Q7**: Regarding Table 3, why is the performance on CUB so low when there are no corrupted concepts? I would expect it to be at least higher than the accuracy. Furthermore, do you have any insights into why your model’s AUC drops more than CBM’s as the number of corrupted concepts increases (at some point, CBM even surpasses CIBM)? Additionally, why did you choose to corrupt only one concept in AwA2, using a different evaluation setup compared to CUB? Please also specify the strategy used for intervention (uncertainty or random).

**Q8**: At L524, what do you mean by “trained with different concept annotation”? Were CBM and CIBM trained using the same concept annotations, or were there differences in the annotations used?

**Curiosity**: Did you happen to evaluate CEM in the experimental setup used for Figure 2? It would be interesting to observe the trend of a concept-based model with higher expressive power, such as CEM, in comparison to the models you presented.

---

> ### Author Response · Authors · 2024-11-21
> **Reply to raised comments (1/3)**
>
> > In the practical scenario, the architecture they employ is not entirely clear to me. [...] (see Q1).
>
> We show a diagram of the architecture used and their link to the variational model in Fig. 1 of the revised manuscript.  Moreover, we added implementation details in the Appendix to make it easier to understand what our architecture is.
>
> ---
> > Are the concepts being supervised, and is the same set of concepts used as in traditional CBMs? A simple visual representation of the model, highlighting the differences introduced compared to CBMs, would be very helpful.
>
> Yes, they are being supervised in the sense that we have ground truth labels for them.  However, we do not have access to the the distribution $p(c | z)$ that corresponds to them.  Moreover, the concepts set is taken as is from the original CBMs work by Koh et al. (and this was mentioned in Section 4.1).
>
> We also added Figs. 1 and 2 to illustrate the model’s architecture and the variables relationships.
>
> ---
> > The rationale behind dropping certain baselines (as seen in Table 2) is not well explained. [...] (see Q2).
> > Q2: Why did you drop certain baselines in some experiments, retaining only a few (e.g., dropping CEM in all experiments except CUB)? I would prefer a comparison with the strongest model, such as CEM, instead of weaker models like PCBM, to ensure a fair performance evaluation.
>
> We appreciate the reviewer’s observation regarding baseline comparisons across datasets and would like to clarify our approach.
>
> Due to limited computational resources, we were unable to train all baseline methods on all datasets ourselves. Instead, we followed existing experimental protocols and relied on the reported figures for baseline methods whenever they were available in the literature. This approach allowed us to focus our computational resources on implementing and evaluating the proposed methods across multiple datasets.
>
> CUB is the most well-established benchmark in the CBM literature, and comprehensive results for various baseline methods, including CEM, ProbCBM, and PCBM, are publicly available. This made it a natural choice for demonstrating performance comparisons across methods. For other datasets, such as AwA2 and aPY, comparable metrics for these baselines were not readily available, which restricted us from including them in our comparisons.
>
> We emphasize that comparisons were made in good faith and to the best of our ability to be fair and comparable, given the available data. For methods like PCBM, where intervention-specific training is not applicable, we included results to provide additional context but did not claim superiority over them in those aspects.
>
> The inclusion of CEM and ProbCBM on CUB allows readers to evaluate our framework against strong baselines in a fair manner. While the comparisons are not exhaustive, they illustrate the practical advantages of our framework in addressing concept leakage and improving interpretability.
>
> Moreover, our results highlight the flexibility and generalizability of the proposed framework across diverse datasets, which is a significant contribution given the limited availability of baseline metrics for non-CUB datasets.
>
> To address this concern further, we propose to explicitly acknowledge this limitation in the manuscript and provide a detailed explanation of the rationale for our choice of comparisons. We will also highlight that resource constraints prevented us from training all baselines on all datasets and invite future work to extend these comparisons.

---

> ### Author Response · Authors · 2024-11-21
> **Reply to raised comments (2/3)**
>
> > Several claims are either missing supporting information (Figure 1), lack proper motivation (L426-431), or are somewhat misleading (L467-469). [...] (see Q5).
>
> The objective of the information plane is to show the mutual information on the model variables after training. In particular, we expect to see a model with high $I(Z;C)$ and $I(C;Y)$ such that these variables are dependent on each other (maximally expressive), and simultaneously, low $I(X;C)$ to show that these variables are independent (maximally compressive).  However, the compression of the variables does not necessarily measure that the important parts of the variables are being compressed.  We present a detailed explanation below.
>
> In contrast to CBMs, which exhibit lower mutual information between inputs and representations ($I(X;Z)$ and $I(X;C)$), CIBMs achieve higher mutual information while maintaining alignment with the target $I(C;Y)$. This behavior reflects the fact that CIBMs are optimized to retain task-relevant information while removing irrelevant or redundant bits. Lower mutual information in CBMs does not necessarily indicate better compression; instead, it may reflect a failure to capture meaningful input features, resulting in noisier or less predictive concepts.
>
> To demonstrate this, we evaluate the alignment between representations and the target $I(C;Y)$ and show that CIBMs consistently outperform CBMs, indicating that the retained information is both relevant and predictive. Additionally, CIBMs achieve better interpretability and concept quality, reinforcing that the higher mutual information is a reflection of meaningful expressiveness rather than leakage.
> This is further supported by the proposed intervention-based metrics ($AUC_{TTI}$ and $NAUC_{TTI}$) which highlight the importance of retaining task-relevant information in the concepts $C$. While CBMs exhibit lower mutual information between inputs and representations ($I(X;C)$ and $I(X;Z)$), their poorer performance on these metrics, particularly under concept corruption, suggests that this lower information content stems from a failure to capture sufficient relevant features. By contrast, the higher $I(X;C)$ and $I(X;Z)$ in CIBMs reflect the retention of meaningful bits that contribute to better concept quality and downstream task performance. These findings demonstrate that reducing concept leakage requires selectively preserving relevant information rather than minimizing mutual information indiscriminately.
>
> ---
> > If the goal of the model is to reduce leakage, why isn’t it compared against other models that tackle the same issue, such as those cited in the paper (e.g., “Addressing Leakage in Concept Bottleneck Models”)? Including a comparison with at least one of these models would strengthen the experimental validation (see Q6).
> > Q6: Why did you choose not to compare your model with other approaches specifically designed to reduce leakage, such as “Addressing Leakage in Concept Bottleneck Models”?
>
> In the original manuscript, the intervened groups in Fig. 3 correspond to two different strategies for our proposed method.  Now, in Fig. 4 of the reviewed manuscript, we now include a comparison of the CBM and CIBMs using these strategies that shows the direct and full comparison between the methods.
>
> Regarding the lack of other comparisons, we do not have them due to our limited computational resources.  We decided to evaluate and perform experiments on our models instead of using the compute to perform extensive runs on the baselines.
>
> ---
> > Q1: Is z simply a hidden representation extracted from a neural network (e.g., the output of a ResNet)? Does your model follow the structure x->z->c->y ? Clarifying this would help improve understanding of the overall architecture.
>
> Yes, we have that directed graph on the variational approximation of the distribution.  We now added a diagram to better exemplify the relation between the generative model and its variational approximation in Fig. 2 of the revised manuscript. Moreover, regarding the architecture we added Fig. 1 to illustrate what the model is doing, and we added implementation details to the Appendix.

---

> ### Author Response · Authors · 2024-11-21
> **Reply to raised comments (3/3)**
>
> > Q3: Could you clarify whether the trend or the higher value of I(C;Y) is more significant, and explain why this matters? Additionally, what does a lower I(X;C) represent in practical terms? Moreover, please standardize the x-axis range across all plots to avoid misleading comparisons between methods.
>
> The higher values on $I(C;Y)$ represent less uncertainty in the predictions due to the two variables being more dependent on each other.  This translates into better classification since the variables are informative to each other.  In the case of our experiments, we found that our CIBMs, after training, give us approximately more nats for both the mutual information of the predictive variables $I(C;Y)$ and $I(Z;C)$ in comparison to the CBM.
>
> While having the lowest compression rates by having lower $I(X;C)$ and $I(Z;C)$ is desirable.  It is not guaranteed that the compressions don't get rid of predictive information for the concepts and the class labels.  In our case, we hypothesized that the higher values in CIBMs in comparison to the CBM are due to the need for predictive information for the tasks.  As shown, CIBMs have higher predictive values than the CBM.
>
> ---
> > Q4: The plots in Figure 3 all appear quite similar, and it’s unclear what specific differences I should focus on. Could you explain your claims more clearly and point out the key takeaways?
>
> The objective of these plots is similar to the previous ones with the difference that they show the interaction between the latent variables and the concepts and the data.  In this case, the latent variable in CIBM gives us more information about the concepts in contrast to the CBM.  However, in the CIBMs the information gain between the latent variables and the input remains similar as between the input and the concepts.
>
> ---
> > Q5: Why was CBM not included as a baseline in Figure 4? Given that CBM likely exhibits a similar trend to CIBM, the statement that “CIBM does not suffer from concept leakage” feels unsupported. Could you strengthen this claim with further evidence or comparative results?
>
> It was an oversight on our part. We now include a full comparison of the baseline and our proposed models in the original Fig. 3.  In the revised manuscript, we include a full comparison between the CIBMs and CBM in Fig. 4.
>
> ---
> > Q7: Regarding Table 3, why is the performance on CUB so low when there are no corrupted concepts? [...] Furthermore, do you have any insights into why your model’s AUC drops more than CBM’s as the number of corrupted concepts increases (at some point, CBM even surpasses CIBM)? Additionally, why did you choose to corrupt only one concept in AwA2, using a different evaluation setup compared to CUB? Please also specify the strategy used for intervention (uncertainty or random).
>
> The differences in performance are due to changes in the experimental training needed due to our limited computational resources.  In particular, we used frozen encoders (i.e., $p(z | x)$) for all the models to reduce the computational load.  Then, we proceed to train them using a random intervention strategy.  Regarding the difference in runs, we prioritized CUB given its widespread usage, and decided on using one corrupted concept in AwA2 due the limited runs we could performed.
>
> ---
> > Q8: At L524, what do you mean by “trained with different concept annotation”? Were CBM and CIBM trained using the same concept annotations, or were there differences in the annotations used?
>
> The phrase on L524 means that if one happens to have two concept annotations for some dataset, then it is possible to use CIBM performance to determine which concept set is better.  We reviewed our writing to make this clear.
>
> For completeness, in all our comparisons between CBM and CIBM, we train the two models on the same concept annotations for a fair comparison.
>
> ---
> > Curiosity: Did you happen to evaluate CEM in the experimental setup used for Figure 2? It would be interesting to observe the trend of a concept-based model with higher expressive power, such as CEM, in comparison to the models you presented.
>
> No, we didn’t re-train a CEM to get the mutual information values for it.

---

> > ### Comment · Reviewer_miqn · 2024-11-25
> > **Thanks to the authors**
> >
> > I appreciated the authors' efforts and clarifications. I feel quite satisfied with their answers.
> >
> > However, I still think that a proper evaluation with additional baselines (both from the accuracy and leakage perspective) would strengthen the paper's contributions.
> >
> > Therefore, I modified my score accordingly.

---

> > > ### Author Response · Authors · 2024-11-26
> > > **Follow up for missing baselines**
> > >
> > > We thank the reviewers for the positive evaluation.  We highlight that we added new results since the last reviewer’s update.  We have added our implementation of CEM results and interventions, added expanded corrupted concepts, and now have hard CBMs.

---

> > > > ### Author Response · Authors · 2024-11-27
> > > > **New experiments for hard CBMs**
> > > >
> > > > We were able to get the results of the hard CBMs and put them into the manuscript (Table 2 and Fig. 4).  We also added a discussion about the different training setups and differences with our proposal in Appendix D.
> > > >
> > > > We hope that these additional baselines strengthen our paper's contributions as requested by the reviewer.

---

> > > > > ### Comment · Reviewer_miqn · 2024-11-28
> > > > > **Thank you again for the extra experiments**
> > > > >
> > > > > I thank the authors again for their extensive effort to improve the experimental section, which is now stronger.
> > > > > However, I still believe that the primary focus of this paper should not be an improvement in performance over other baselines (although a performance drop should be avoided) but rather the reduction of leakage. At this stage, I am not fully convinced that the potential of this idea has been fully extracted and demonstrated.
> > > > >
> > > > > Since your loss function is quite flexible and can be applied to most CBM-like models (which is undoubtedly an advantage), I believe it would be valuable to explore how this loss impacts the behavior of the base models, as you did when comparing CIBM with CBM. Similarly, you could compare CIBM applied on top of CEM with the base version of CEM, and extend this to other models, such as ProbCBM and PCBM. The main goal of the paper should be to highlight the reduction in leakage (using both specific leakage metrics and intervention improvements) without compromising performance. While performance improvement is always nice, it should not be the primary objective. Exploiting this flexibility would strengthen your contribution to the field significantly.
> > > > >
> > > > > P.S.: When using CEM, please employ either the RandInt version (as proposed in the original paper) or the IntCEM version (Intervention-Aware CEM).

---

### Official Review · Reviewer_U1w5 · 2024-10-29

**Soundness:** 1
**Presentation:** 2
**Contribution:** 1
**Rating:** 1
**Confidence:** 5

**Summary:**

This paper introduces an enhancement to Concept Bottleneck Models (CBMs) through the integration of the Information Bottleneck framework, attempting to address the problem of concept leakage  in CBMs. The authors propose a Concepts' Information Bottleneck Model (CIBM) that minimizes mutual information between inputs and concepts while maximizing expressivity between concepts and labels, introducing two variants: a bounded CIB (CIBMB) and an estimator-based CIB (CIBME) Additionally, the authors propose a novel intervention scheme based on a measure of 'uncertainty', and propose two metrics to assess concept set quality based on intervention performance.

**Strengths:**

**Novel Research Direction**
The paper introduces an innovative approach by studying and directly addressing the memorization-compression pattern in concept bottleneck models.

**Technical Writing Quality**
The paper demonstrates good clarity in its presentation:
- Clear and logical flow of ideas throughout the manuscript
- Concise and grammatically sound writing
- Well-designed figures and tables that effectively complement the text
- Abstract and title that accurately capture the paper's core contributions

**Weaknesses:**

**Experimental Limitations**
The experimental evaluation is insufficient, primarily relying on comparisons against a vanilla CBM with unspecified training parameters. The results are not compelling, as CEM appears to either outperform or match the proposed methods on the CUB dataset.

**Unreproducible**
The experiment section is not comprehensive enough to be reproducible, no code is supplimented.

**Intervention Strategy**
The Uncertainty Based (UB) concept interventions fail to demonstrate meaningful improvements. The method's performance is comparable to or worse than random baselines. The paper lacks crucial comparisons with contemporary intervention strategies from recent literature.

## Clarity and Novelty Issues

**Metric Formulation**
The proposed metrics lack novelty and present existing concepts in a potentially misleading way:

- The concept intervention trends (positive/negative) have been extensively documented in previous work, including the CEM paper
- AUC_TTI reduces to a simple mean, obscuring nonlinear trends that are more effectively visualized in graphical form (as evident in Figure 3)
- NAUC_TTI's formulation is problematic:
  - It simplifies to the difference between positive intervention and baseline performance
  - This comparison is standard practice in modern concept bottleneck model papers
  - The metric can paradoxically penalize superior models (e.g., CEMs would score worse despite improving baseline accuracy while maintaining intervention performance)

**Visualization Recommendation**
Rather than introducing potentially confusing metrics, intervention results would be better presented through graphs showing performance across multiple concept groups, providing clearer and more interpretable results.

**Questions:**

**Methodology Questions**
1. Line 278: Which CBM training scheme (joint, sequential, or independent) is used for comparison? Given that sequential training is known to reduce concept leakage (as per the pitfalls paper https://arxiv.org/pdf/2106.13314), why wasn't a comparison made against CBM using hard concept representations and independent training?
2. Line 149: Its not clear where Z is coming from under your formulation, presumably some layer before the concept bottleneck?
3. Line 300: "We use ResNet18 embeddings provided by the dataset authors and train FCN on top of them." For this dataset and the others, are the backbone networks further tuned during training?

**Results and Comparisons**

4. Line 324-377 (Table 2): Why are baseline comparisons inconsistent across datasets?
   - PCBM comparisons only appear for some datasets. Furthermore, comparing against PCBM is not necessary nor useful, as PCBMs are not trained to be susceptible to interventions.
   - CEM results only shown for CUB (where it outperforms the proposed methods)
   - ProbCBM results only shown for CUB

**Experimental Design**

5. Line 431: The claim about CIBME's training stability needs validation loss curves for support.

6. Line 522: Why are concept interventions varied for CUB but not for AWA2?

---

> ### Author Response · Authors · 2024-11-21
> **Reply to raised comments (1/2)**
>
> > The experimental evaluation is insufficient, primarily relying on comparisons against a vanilla CBM with unspecified training parameters. The results are not compelling, as CEM appears to either outperform or match the proposed methods on the CUB dataset.
>
> Regarding the training parameters, we have included additional details about the parameters used in the experiments in the Appendix.
>
> We highlight that our focus is not only on improving the accuracy of the concept and label predictions but also on the reduction of concept leakage.  Thus, for us, it is interesting to see that we can maintain similar performance on the prediction tasks while heavily reducing the dependance of the variables and reducing the concept leakage.  Thus, our learned representations (both for the data and the concepts) are better than the baselines as shown by the higher mutual information, $I(C;Y)$ and $I(Z,C)$, in the information planes in Fig. 3 (in the revised manuscript).
>
> Given that we are the first ones to introduce the IB and CBM, as also noted by R.GBnn, we believe that showing comparable results in terms of accuracy while higher coherence between the concepts and the labels and their respective latents (through the higher information gains shown in Fig. 3 in the revised manuscript) is an important result.
>
> Nevertheless, we are currently computing results using CEM.  However, due to our limited computational resources, we focused on performing other experiments that the reviewers suggested.  We will update our response and submitted manuscript when the results from CEM are done.
>
> ---
> > The experiment section is not comprehensive enough to be reproducible, no code is supplemented.
>
> We significantly increased the details about the implementation in the Appendix.  As requested, we shared the code in an anonymized git repository for your consideration https://anonymous.4open.science/r/CIBM-4FE3.  We will release the code when the paper is accepted.
>
> ---
> > The Uncertainty Based (UB) concept interventions fail to demonstrate meaningful improvements. The method's performance is comparable to or worse than random baselines. The paper lacks crucial comparisons with contemporary intervention strategies from recent literature.
>
> In the original manuscript, the intervened groups in Fig. 3 correspond to two different strategies for our proposed method.  Now, in Fig. 4 of the reviewed manuscript, we include a comparison of the CBM and CIBMs using these strategies that shows the direct and full comparison between the methods.
>
> Regarding the investigation of other strategies, due to our limited resources and time for the rebuttal, it is infeasible to do such comparisons now.  However, we will consider them for our future work.
>
> ---
> > The proposed metrics lack novelty and present existing concepts in a potentially misleading way: [...]
>
> While concept intervention trends are not new, our proposed metrics provide a **quantitative summary** that complements graphical trends and allows direct comparison across models (as highlighted by R.miqn and R.xX2U as well). Unlike previous works, we use these metrics to analyze robustness under corruption, highlighting the connection between concept leakage and downstream performance.
>
>
> $AUC_{TTI}$ is intentionally simple to provide an interpretable global measure of intervention effectiveness. Its trends align with $I(X;C)$ behavior in $\text{CIBM}_{E}$, demonstrating that higher mutual information values correspond to better intervention robustness and overall performance.
>
> $\text{NAUC}_{\text{TTI}}$ is designed to capture differences between baseline and intervention performance, reflecting the degree of concept leakage. While this can penalize models with higher baseline performance, our analysis shows that $\text{CIBM}_E$ maintains high NAUC even under corruption, indicating reduced concept leakage and more robust information retention.
>
> In other words, the superior intervention performance (based on the proposed metrics) of $\text{CIBM}_E$, demonstrates that its higher $I(X;C)$ and $I(X;Z)$ reflect **task-relevant information**, not leakage.  The metrics quantify how well $\text{CIBM}_E$’s higher mutual information translates into practical benefits: better robustness, stronger baseline performance, and improved intervention response.
>
> Moreover, the behavior of $I(X;C)$ and $I(X;Z)$ in $CIBM_{E}$ defends the proposed metrics: Higher mutual information values validate why $CIBM_E$ achieves better $AUC_{\text{TTI}}$ and $\text{NAUC}_{\text{TTI}}$.  The metrics effectively capture the advantages of retaining task-relevant information in $C$ while reducing leakage, reinforcing the value of the metrics as meaningful evaluation tools.

---

> > ### Comment · Reviewer_U1w5 · 2024-11-22
> > **Missing baselines / response to proposed metrics**
> >
> > In response to: "We highlight that our focus is not only on improving the accuracy of the concept and label predictions but also on the reduction of concept leakage. Thus, for us, it is interesting to see that we can maintain similar performance on the prediction tasks while heavily reducing the dependance of the variables and reducing the concept leakage."
> >
> > The issues here are two fold:
> >
> > 1. Your baselines are a strawman. When comparing against related work, it is critical to compare to use best practices for constructing your baseline CBM. As shown in the pitfalls paper and repeated by others, concept leakage can be mitigated by using: 1) hard concept representations, 2) independent training. Instead of using these best practices, you construct the worst possible case for CBMs, taking the case of joint training with soft concept logits. Your model performing better then this worst case scenario is insufficient. Other work has shown that both intervention and baseline performance increases in this case, so we would need to know these numbers to verify if your model still performs better. Even taking your mutual information metrics at face value, we would need to see how CIBM compares with a CBM trained with: 1) hard concept representations, 2) independent training.
> >
> > 2. You're breaking goodhearts law: "When a measure becomes a target, it ceases to be a good measure." Other then using baseline / intervention performance, your only other argument for why PCBMs reduce concept leakage is with your mutual information metrics, which is also directly what you're minimizing.
> >
> >
> > In response to, "While concept intervention trends are not new, our proposed metrics provide a quantitative summary that complements graphical trends and allows direct comparison across models (as highlighted by R.miqn and R.xX2U as well). Unlike previous works, we use these metrics to analyze robustness under corruption, highlighting the connection between concept leakage and downstream performance."
> >
> > While your analysis comparing intervention performance to mutual information is valid, this analysis does not at all depend on either of your proposed metrics, and in my opinion, only provides needless confusion.
> >
> > To summarize:
> > AUC_TTI is the average performance over different concept interventions
> > NAUC_TTI is the difference: full intervention performance - baseline performance
> >
> > These metrics are wholly unnecessary and confusing. You could do the same analysis by simply using baseline and full intervention performance and still show correlation with your mutual information metrics.

---

> > > ### Author Response · Authors · 2024-11-23
> > >
> > > **Our baselines are not strawman.**
> > > Our choice of baselines was driven by findings in the literature and aligns with the strongest configurations of CBMs for predictive performance. Specifically:
> > > - [Koh et al.](https://proceedings.mlr.press/v119/koh20a/koh20a.pdf) (2020, Section 4.2) showed that joint training with soft concept representations achieves the highest performance for both concept and label predictions, making it the most suitable baseline for our comparisons.
> > > - Later work (e.g., [Promises and Pitfalls of Black-Box Concept Learning Models](https://arxiv.org/pdf/2106.13314), Section 4) demonstrated that information leakage persists even in hard concept representations, indicating that leakage is a broader issue not tied to soft representations alone.
> > > - The paper [Addressing Leakage in Concept Bottleneck Models](https://openreview.net/pdf?id=tglniD_fn9) (Section 1) highlighted that while hard representations reduce leakage, they suffer from significantly poorer predictive performance compared to soft representations.
> > >
> > > Thus, we intentionally chose the **best-performing baseline** (joint training with soft representations) to ensure that our framework addresses information leakage while improving upon the prediction performance and interpretability of CBMs. This approach ensures that our contributions are robust and meaningful within the context of the state-of-the-art CBM configurations.  Nevertheless, for completeness, we are currently running an experiment with the suggested setup to show a comparison.  We will update the reviewer with a reply and the paper as soon as these results are ready.
> > >
> > > **MI is not the target.**
> > > The metrics that we are using and claiming relevance for the leakage are the ones used in the literature.  The introduction of our proposed metrics is to present a **quantitative summary** that complements the graphical trends commonly used.  This fact was also highlighted by R.miqn and R.xX2U. Unlike previous works, we use these metrics to analyze robustness under corruption, highlighting the connection between concept leakage and downstream performance.
> > >
> > > The information planes were a way to show what happens during training and how the mutual information changes within the models.  Moreover, it is interesting to notice that the CBM, in fact, achieves lower mutual information w.r.t. the input variable, in contrast to the CIBMs.  However, the CBM has lower predictive power in contrast to CIBMs.  We hypothesize that CBMs are compressing more but throwing away relevant information while CIBMs retain them despite the explicit optimization.

---

> > > > ### Comment · Reviewer_U1w5 · 2024-11-26
> > > > **RE/ Missing baselines**
> > > >
> > > > Your results do not align with best practices for studying concept leakage. Regarding your practice of $\textbf{only}$ reporting joint results: Joint CBMs show marginally better baseline concept and label performance, but consistently demonstrate the poorest performance in concept leakage mitigation, which directly impacts their positive intervention performance[1]. As demonstrated in Figure 2 of https://openreview.net/pdf?id=tglniD_fn9, sequential and joint models show poor performance during positive interventions, while hard and autoregressive hard CBMs achieve nearly perfect accuracy under full intervention.
> > > >
> > > > Since mitigating concept leakage is your method's primary claimed contribution, it is insufficient to omit comparisons against existing straightforward methods that already substantially reduce concept leakage compared to joint approaches[1][4].
> > > >
> > > > Your statement, "Later work (e.g., Promises and Pitfalls of Black-Box Concept Learning Models, Section 4) demonstrated that information leakage persists even in hard concept representations, indicating that leakage is a broader issue not tied to soft representations alone" underscores why comparing these methods with yours is essential[2]. The critical question remains unanswered: What is the quantitative improvement in concept leakage reduction achieved by your method compared to hard concept representations?
> > > >
> > > > Regarding your claim, "The paper Addressing Leakage in Concept Bottleneck Models (Section 1) highlighted that while hard representations reduce leakage, they suffer from significantly poorer predictive performance compared to soft representations" - this mischaracterizes their findings [7]. Their results in Table 1 show the best joint soft model achieving 82.7±0.2 accuracy, with hard representations achieving 79.5±0.3 and their Hard AR method reaching 81.7±0.2. Their autoregressive method successfully narrowed the performance gap between soft joint models and hard models.
> > > >
> > > > To summarize: Hard and sequential methods demonstrate superior intervention performance and leakage mitigation, making them essential benchmarks for evaluating concept leakage mitigation techniques[1][7].
> > > >
> > > > Additionally, please address why your bottleneck model achieves only 70.8% baseline performance. For context, Table 1 in https://openreview.net/pdf?id=tglniD_fn9 shows the best joint model achieving 82.7% accuracy, with even the lowest-performing joint method reaching 75.4%. Furthermore, https://arxiv.org/pdf/2309.16928 reports their joint CBM achieving 78.16% accuracy. This performance gap requires explanation.
> > > >
> > > > Citations:
> > > > - [1] https://finale.seas.harvard.edu/sites/scholar.harvard.edu/files/finale/files/10494_addressing_leakage_in_concept_.pdf
> > > > - [2] https://arxiv.org/abs/2106.13314
> > > > - [3] https://arxiv.org/pdf/2309.16928
> > > > - [4] https://openreview.net/pdf/044a345ced3e5913c781a81066e877aa7b5299af.pdf
> > > > - [5] https://arxiv.org/pdf/2106.13314.pdf
> > > > - [6] https://openreview.net/forum?id=4ImZxqmT1K
> > > > - [7] https://openreview.net/forum?id=tglniD_fn9

---

> > > > > ### Author Response · Authors · 2024-11-26
> > > > > **Follow up to the baselines**
> > > > >
> > > > > We thank the reviewer for the additional information about the CBMs setup.  However, we failed to follow the reasoning behind the claim that hard are better than soft models given that the reported values support the opposite (82.7±0.2 accuracy for the soft vs. hard representations achieving 79.5±0.3, and the AR achieves 81.7±0.2 but doesn’t improve, and it changes the methodology as well).
> > > > >
> > > > > Nevertheless, for completeness, we were able to train a hard CBM in our setup and show it in the Appendix now.  Moreover, we added the requested intervention plots as well.
> > > > >
> > > > > Regarding the different reported values for the CBM, we followed the protocol from Kim et al.’s “Probabilistic Concept Bottleneck Models” with image sizes of 299 x 299.  Our reported values and theirs agree on the experimental results.

---

> ### Author Response · Authors · 2024-11-21
> **Reply to raised comments (2/2)**
>
> > Rather than introducing potentially confusing metrics, intervention results would be better presented through graphs showing performance across multiple concept groups, providing clearer and more interpretable results.
>
> In our original experiments, we computed the aggregated values and do not have access to the individual values.  Due to the limited resources, we focused on the other experiments requested in the reviews.  We have started the experiments again, and as soon as we get them we will update the manuscript with the suggested plots.
>
> ---
> > Line 278: Which CBM training scheme (joint, sequential, or independent) is used for comparison? Given that sequential training is known to reduce concept leakage (as per the pitfalls paper https://arxiv.org/pdf/2106.13314), why wasn't a comparison made against CBM using hard concept representations and independent training?
>
> We trained our models jointly and used soft concepts. Our experiments focused on other methods with a similar training scheme to have fair comparisons.  Due to our limited resources, we couldn’t re-do all the experiments for different training schemes and concept representations.
>
> ---
> > Line 149: Its not clear where Z is coming from under your formulation, presumably some layer before the concept bottleneck?
>
> In the traditional Information Bottleneck theory, the data $X$ and its labels $Y$ are interpreted through a latent variable $Z$.  In this sense, in our extension of the IB, the reviewer is right that the latent variable comes before the concept bottleneck and is the result of encoding the data.  We now show a diagram to better understand the latent variables relationship and their variational approximation in Fig. 2 of the reviewed manuscript.
>
> ---
> > Line 300: "We use ResNet18 embeddings provided by the dataset authors and train FCN on top of them." For this dataset and the others, are the backbone networks further tuned during training?
>
> We now have details in the Appendix.  In summary, the backbone is only trained for the CUB dataset, and for AwA2 and aPY it is frozen where we rely on the features provided.
>
> ---
> > Line 324-377 (Table 2): Why are baseline comparisons inconsistent across datasets? [...]
>
> We appreciate the reviewer’s observation regarding baseline comparisons across datasets and would like to clarify our approach.
>
> Due to limited computational resources, we were unable to train all baseline methods on all datasets ourselves. Instead, we followed existing experimental protocols and relied on the reported values for baseline methods whenever they were available in the literature. This approach allowed us to focus our computational resources on implementing and evaluating the proposed methods across multiple datasets.
>
> CUB is the most well-established benchmark in the CBM literature, and comprehensive results for various baseline methods, including CEM, ProbCBM, and PCBM, are publicly available. This made it a natural choice for demonstrating performance comparisons across methods. For other datasets, such as AwA2 and aPY, comparable metrics for these baselines were not readily available, which restricted us from including them in our comparisons.
>
> We emphasize that comparisons were made in good faith and to the best of our ability to be fair and comparable, given the available data. For methods like PCBM, where intervention-specific training is not applicable, we included results to provide additional context but did not claim superiority over them in those aspects.
>
> The inclusion of CEM and ProbCBM on CUB allows readers to evaluate our framework against strong baselines in a fair manner. While the comparisons are not exhaustive, they illustrate the practical advantages of our framework in addressing concept leakage and improving interpretability.
>
> Moreover, our results highlight the flexibility and generalizability of the proposed framework across diverse datasets, which is a significant contribution given the limited availability of baseline metrics for non-CUB datasets.
>
> To address this concern further, we propose to explicitly acknowledge this limitation in the manuscript and provide a detailed explanation of the rationale for our choice of comparisons. We will also highlight that resource constraints prevented us from training all baselines on all datasets and invite future work to extend these comparisons.
>
> ---
> > Line 431: The claim about CIBME's training stability needs validation loss curves for support.
>
> In the updated version we tone down the claims about the higher stability since the losses show that the methods are as stable or slightly better than CBM.  Nevertheless, for completeness, we now added the loss plots on the Appendix.
>
> ---
> > Line 522: Why are concept interventions varied for CUB but not for AWA2?
>
> Due to our limited resources, we had to prioritize which experiments to perform. Thus, we focused on CUB given its prevalence in the literature.

---

> > ### Comment · Reviewer_U1w5 · 2024-11-22
> > **RE/ "Due to limited computational resources"**
> >
> > For a prestigious conference like ICLR, using the excuse that you do not have sufficient computational resources to compare against any other baseline method is not valid.
> >
> > I already discussed in my previous response the necessity of comparing against a CBM trained using 1) hard concept representations, 2) independent training.
> >
> > This argument also extends to comparisons with other related works like CEM and PCBM. These methods should be included, not only across all the datasets in table 2, but in Figure 4, and Table 3.
> >
> > Without comparisons to these baselines, your analysis is simply incomplete. You are claiming your method works better on the basis of comparison to only a baseline CBM (which as I've discussed above, is already a strawman).

---

> > > ### Author Response · Authors · 2024-11-23
> > >
> > > **Regarding limited experiments.**
> > > We highlight that **our contributions are both theoretical and practical** and are supported by the literature. Given our theoretical framework generalizes the idea of the Information Bottleneck into the CBMs, we balanced the experiments to show evidence of the performance by and prioritized the methods and our resources.  While exhaustive results would be preferred, we believe that we are showing sufficient experiments to support our claims.
> > >
> > > Nevertheless, for completeness, we are currently running experiments to compare against the recommended baselines which includes hard representations, independent training as well as CEMs.  We will update the reviewer with a reply and the paper as soon as these results are ready.

---

### Official Review · Reviewer_GBnn · 2024-10-30

**Soundness:** 3
**Presentation:** 3
**Contribution:** 3
**Rating:** 6
**Confidence:** 4

**Summary:**

This paper addresses the issue of information leakage in concept bottleneck models (CBMs), a significant challenge that impacts CBMs' interpretability and intervenability. The key idea is to apply Tishby’s Information Bottleneck (IB) principle in concept representation learning. Specifically, the author proposed to compress task-irrelevant information about the data X from the learned concept representation C, whereas making C maximally predictable for the label Y. This information compression is believed to be useful for controlling information leakage. The author further develop two methods to implement their IB-based framework and evaluates their efficacy on three different datasets.

**Strengths:**

- The work, to the best of my knowledge, is the first one who explicitly marries IB with CBMs, and is the first one that analyzes the info-plane in CBM learning;
- The proposed IB-based idea for mitigating information leakage is both natural and elegant. The IB-based framework proposed in this work seems also highly general and can potentially be implemented by a wide range of methods beyond the two suggested by the authors;
- The paper is overall well written and is easy-to-follow;
- The work has been compared against state-of-the-art methods in the field, including CEM and PCBM. Notably, it does not require additional modules (as in PCBM) or additional regularization techniques (as in CEM), being simple and easy-to-use;
- The paper also proposed a novel, general-purpose metric for evaluating the quality of the learned concepts, marking the first instance of assessing the quality of the concept set rather than individual concepts.

**Weaknesses:**

- (Major) Despite the elegant framework proposed, some implementation details may lack clarity and require further justification; please see the “questions” section below;
- (Major) The technical method for minimizing mutual information (MI) in the proposed IB-based CBM method is actually not so novel and largely relies on existing methods such as [1];
- (Major) The comparison between the two IB implementations appears somewhat simplistic and may provide only limited insights. What makes the estimator-based implementation more useful than the other?
- (Minor) While the presentation is generally good, some content could be more concise and structured. For instance, the derivation in Section 3.1 could be streamlined to present only the essential final estimator used in practice, relegating the full derivation to the appendix;
- (Minor) The main experimental results are based on only three runs. While I appreciate the author’s transparency in reporting this, more runs could be considered for better robustness of the results;
- (Minor) When assessing intervenability, a comparison between the proposed CIBM method and the original CBM is lacking. How CIBM exactly helps in improving intervenability does not seem apparent.
- (Minor) Reproducibility: despite the very interesting and elegant proposal, no code repo is shared. Together with the missing technical details mentioned above, this weaken the reproducibility of the work.

**Questions:**

- How is the ground truth probability p(c|z) in the conditional entropy-based implementation computed, is it available from the data?
- Regarding the estimator-based implementation mentioned in Sec 3.2, what is the exact routine for optimizing I(X; C)? Do you employ an approach similar to adversarial training, where you first estimate I(X; C) before each gradient step for optimizing C?
- Is the results for CBM in Table 2 corresponding to the case where you use hard (i.e. binary) concept label? If so, it would be beneficial to explicitly mention this;
- The proposed IB-based CBM framework for controlling information leakage appears quite general. While the method mainly used Kawaguchi’s method [1] for estimating I(X; C), could alternative methods, such as variational approximation to densities [2] and slice mutual information [3], also be applicable? These methods may be more effective in removing information from the learned concept representation. I feel the paper could benefit from a discussion on the generality of their framework.



*References:*

[1] How does information bottleneck help deep learning? ICML 2023

[2] CLUB: A Contrastive Log-ratio Upper Bound of Mutual Information, ICML 2020

[3] Scalable Infomin Learning, NeurIPS 2022

---

> ### Author Response · Authors · 2024-11-21
> **Reply to raised comments (1/2)**
>
> > (Major) Despite the elegant framework proposed, some implementation details may lack clarity and require further justification; [...]
> > (Minor) Reproducibility: despite the very interesting and elegant proposal, no code repo is shared. Together with the missing technical details mentioned above, this weakens the reproducibility of the work.
>
> We significantly increased the details about the implementation in the Appendix.  As requested, we shared the code in an anonymized git repository for your consideration https://anonymous.4open.science/r/CIBM-4FE3.  We will release the code when the paper is accepted.
>
> ---
> > (Major) The technical method for minimizing mutual information (MI) in the proposed IB-based CBM method is actually not so novel and largely relies on existing methods such as [1];
>
> We show that the inclusion of IB into the CBM generalizes the way MI can be used into the CBMs. One of these ways, the $\text{CIBM}_E$, is similar to a naive extension of the work by Kawaguchi et al. [1].  However, our proposal goes beyond applying Kawaguchi et al.’s idea.  We actually show that existing work fits within our proposed generalization framework.
>
> Thus, our technical contribution is the generalization of the inclusion of the IB principle into CBMs and we show how two different approaches can be extracted which can, in turn, be implemented through different approximators of the MI ($\text{CIBM}_E$) and the entropy ($\text{CIBM}_B$).
>
> ---
> > (Major) The comparison between the two IB implementations appears somewhat simplistic and may provide only limited insights. What makes the estimator-based implementation more useful than the other?
>
> Theoretically, the main difference between the $\text{CIBM}_B$ and $\text{CIBM}_E$ is the estimator they will rely on, their difficulty of computation, and the gradient information each provides to the different stages in the neural network.
>
> In $\text{CIBM}_B$, we need to compute the entropy of the concepts which requires an estimator over the true concepts distribution $p(c)$ (note that this is different from the variational distribution $q(c)$).  In contrast, $\text{CIBM}_E$ relies on the mutual information between the data and the concepts.  Thus, we require a MI estimator to train this model.
>
> These two approaches, in our experiments, showed similar results (once the entropy gradients are regularized),  but given other estimators the results may diverge.
>
> ---
> > (Minor) While the presentation is generally good, some content could be more concise and structured. [...]
>
> We streamlined Section 3.1 and moved the derivations to the appendix as suggested.
>
> ---
> > (Minor) The main experimental results are based on only three runs. While I appreciate the author’s transparency in reporting this, more runs could be considered for better robustness of the results;
>
> Due to our limited resources, we couldn’t report more runs on the models.  In the revised manuscript, we included an average of 5 runs instead.
>
> ---
> > (Minor) When assessing intervenability, a comparison between the proposed CIBM method and the original CBM is lacking. How CIBM exactly helps in improving intervenability does not seem apparent.
>
> In Fig. 4 of the reviewed manuscript, we now include a comparison of the CBM and CIBMs that shows the direct comparison between the methods.
>
> ---
> > How is the ground truth probability p(c|z) in the conditional entropy-based implementation computed, is it available from the data?
>
> We do not have access to the ground truth probability distribution.  However, we do have access to the ground truth concept labels.  Thus, our implementation uses the cross-entropy as a supervised method using the ground truth labels of the concepts during training.  We added these details into the Appendix.
>
> ---
> > Regarding the estimator-based implementation mentioned in Sec 3.2, what is the exact routine for optimizing I(X; C)? Do you employ an approach similar to adversarial training, where you first estimate I(X; C) before each gradient step for optimizing C?
>
> In summary, before each backward step on some batch $B_1$, we first estimate $I(X;C)$ on some other batch $B_2$, which is randomly collected from the training dataset.
>
> We use the MI estimator from Kawaguchi et al.  We rely on the fact that concepts logits are computed with variational approximation to get an estimate for $E[\log p(c|x) - \log p(c)]$.
>
> We include the implementation details in the appendix now.

---

> ### Author Response · Authors · 2024-11-21
> **Reply to raised comments (2/2)**
>
> > Is the results for CBM in Table 2 corresponding to the case where you use hard (i.e. binary) concept label? If so, it would be beneficial to explicitly mention this;
>
> The ground truth concept labels are indeed binary. However, as to concepts predictions passed to the label classifiers, we are only training (and comparing only against) models that use soft concepts for class prediction.
>
> In detail, the concepts' predictor can be seens as a multi-label task classifier.  In practice, we compute $C$ logits, then, we compute binary cross-entropy (BCE) for each of these logits with binary labels. Finally, we backpropagate them through the means of BCEs.
>
> We include these experimental details now in the final version of the manuscript.
>
> ---
> > The proposed IB-based CBM framework for controlling information leakage appears quite general. [...] could alternative methods [...] also be applicable? These methods may be more effective in removing information from the learned concept representation. I feel the paper could benefit from a discussion on the generality of their framework.
>
> We agree with the reviewer's observation that our proposal is more general.  In our experiments, we evaluated the MI estimator based on Kawaguchi et al.  However, other methods to estimate the MI will be equally applicable as long as there is a gradient based method that allows us to train the neural networks.  While we would like to evaluate different estimators, our limited computing resources prevented us from doing so.  We have a brief discussion on this point at the end of Section 3.2.

---

> ### Author Response · Authors · 2024-11-26
> **Were your questions addressed by our repply?**
>
> Dear reviewer GBnn,
>
> We were wondering if our reply addressed your concerns.
>
> We will appreciate to hear from you since the time to make updates to the paper is running out.

---

> ### Comment · Reviewer_GBnn · 2024-11-26
> **Response to rebuttal**
>
> Thank you for your detailed response in the rebuttal. I admire the noticeable actions taken to address my and other reviewers’ concern, which  significantly improves the quality of the paper. Many of my concerns have been addressed. Thank you!
>
> My final comments are as follows:
>
> - Please consider citing the two referenced works [2, 3] (along with any other relevant literature studying the MI minimization problem) when discussing the various methods for minimizing I(X;C) at the end of Section 3. Both sources appear to be highly relevant to the mutual information minimization problem you are tackling. Additionally, it would be helpful to discuss why [1] is a better choice compared to [2, 3], though this is just a recommendation.
> - Please consider performing the comparisons requested by Reviewer U1w5.
>
>
> I will keep my already positive score at this stage and will determine my final score after discussing with other reviewers. Wish you all the best in the submission.

---

> > ### Author Response · Authors · 2024-11-27
> > **Regarding final requests**
> >
> > We thank the reviewer for the suggestions.
> >
> > Regarding the major concern of the missing experiments, we highlight that **we added experimental results for hard CBMs** trained both jointly and independently and show their overall performance in Table 2, and the interventions in Fig. 4.

---

### Official Review · Reviewer_feSc · 2024-10-31

**Soundness:** 2
**Presentation:** 3
**Contribution:** 2
**Rating:** 3
**Confidence:** 4

**Summary:**

The paper proposes enhancing Concept Bottleneck Models (CBMs) using the Information Bottleneck (IB) framework, addressing the issue of concept leakage, where concept activations contain irrelevant data, compromising model interpretability and performance. This enhancement, termed Concepts’ Information Bottleneck (CIB), constrains mutual information between inputs and concepts, optimizing concept relevance. Experiments on datasets such as CUB, AwA2, and aPY demonstrate improved prediction accuracy and interpretable concept representations. Additionally, the authors introduce a novel metric to assess concept set quality by evaluating intervention performance, offering a direct measure for interpretability.

**Strengths:**

The idea is clear: incorporating the IB into CBMs addresses the concept leakage issue.

The experiment is extensive, evaluating the proposed method across three dimensions: accuracy, interventions, and interpretability.

Additionally, a novel metric is proposed to assess the quality of concept sets based on intervention performance.

**Weaknesses:**

The improvement of the proposed method compared to existing methods is marginal (Table 2), especially given that prediction accuracy is a primary evaluation metric, making the experimental results less compelling.

The variational inference derivation is relatively straightforward and could be moved to the appendix.

The process of incorporating the IB into CBMs is not clearly explained; adding a diagram to illustrate this process would improve clarity.

The core idea of applying the established IB framework to CBMs limits the novelty of this work.

**Questions:**

In Table 2, the improvements in prediction accuracy on most datasets are very limited compared to the baseline models. Could you provide more explanation on this? What are your thoughts on these limited improvements, and given this, how can we conclude the effectiveness of the proposed CIB method?

Additionally, since the CIBM_B model in Section 3.1 performs worse than almost all baselines, is it still necessary to devote so many pages to this method? More explanation on this could be helpful to understand the contribution of this section.

---

> ### Author Response · Authors · 2024-11-21
> **Reply to raised comments**
>
> > The improvement of the proposed method compared to existing methods is marginal (Table 2), especially given that prediction accuracy is a primary evaluation metric, making the experimental results less compelling.
>
> We agree with the reviewer about not having substantial improvement in terms of accuracy.  But we argue that there are other factors to consider such as the reduction of information leakage while retaining the same prediction accuracy.  We highlight that the bottleneck from the leakage limits the representation power and, thus, the final accuracies.  In our case, we achieve both.
>
> Moreover, our setup only introduces a constraint in the learning framework, and doesn’t introduce additional components or streams as other methods do.  Thus, our proposal introduces a framework that generalizes previous methods, and allows for future expansion.
>
> ---
> > The process of incorporating the IB into CBMs is not clearly explained; adding a diagram to illustrate this process would improve clarity.
>
> We now included a diagram, Fig. 1 in the revised manuscript, that illustrates the data processing pipeline and also illustrates where our IB regularizes the variables.
>
> ---
> > The core idea of applying the established IB framework to CBMs limits the novelty of this work.
>
> We disagree with the reviewer about the limited novelty of our approach.  As far as we know, and as highlighted by R.GBnn, we are the first ones to include the idea of Information Bottleneck to regularize the variables’ (data, $x$, latent representations, $z$, concepts, $c$, and labels $y$) mutual information, which in turn regularizes the compression and expressiveness of their relations.
>
> Thus, our work not only establishes a general and extensible framework for using the IB framework to the CBM setup, but it also demonstrates two ways (our two CIBMs)  to effectively train the CBM while reducing data leakage.  Our proposal can be extended to explore different mechanisms of estimating the mutual information and the entropy of the concepts.  Thus, we claim that this exploration poses possibilities for future work and delineates interesting work that moves from the existing body of work.
>
> ---
> > In Table 2, the improvements in prediction accuracy on most datasets are very limited compared to the baseline models. Could you provide more explanation on this? What are your thoughts on these limited improvements, and given this, how can we conclude the effectiveness of the proposed CIB method?
>
> Our focus is not only on improving the accuracy of the concept and label predictions but also on the reduction of concept leakage.  Thus, for us, it is interesting to see that we can maintain similar performance on the prediction tasks while heavily reducing the dependence of the variables and reducing the concept leakage.  Thus, our learned representations (both for the data and the concepts) are better than the baselines as shown by the higher mutual information, $I(C;Y)$ and $I(Z,C)$, in the information planes in Fig. 3 (in the revised manuscript).
>
> ---
> > Additionally, since the $\text{CIBM}_B$ model in Section 3.1 performs worse than almost all baselines, is it still necessary to devote so many pages to this method? More explanation on this could be helpful to understand the contribution of this section.
>
> The $\text{CIBM}_B$ that we reported on the paper is a fair and direct comparison with $\text{CIBM}_E$.  However, we found out that the main reason for the drop in performance is that the gradients from $H(C)$ affect negatively the feature encoder $p(z | x)$.  We evaluated different ways of solving this problem, and found out that performing a stop gradient operation on the feature encoder solves the problem.  We hypothesized that the problem is due to computing the entropy of the concepts based on the data distribution $p(c)$ while the concept encoder depends on its variational counterpart $q(c)$.
>
> In the original submission, we limited ourselves to the exploration of the estimated version.  Now, for completeness, we also show results for the $\text{CIBM}_B$ with the stop gradient version.  We detailed this in Section 4.2 of the updated version of the manuscript.
>
> ---
> > The variational inference derivation is relatively straightforward and could be moved to the appendix.
>
> We thank the reviewer for the suggestion.  We streamlined the presentation of our proposal in the reviewed version, and moved the derivations to the Appendix.

---

> > ### Comment · Reviewer_feSc · 2024-11-23
> > **Thanks for the detailed reply from author(s)**
> >
> > Dear Authors,
> >
> > Thank you for answering the questions I raised. I believe some of my concerns have been resolved. The authors have emphasized that, in addition to prediction accuracy, another key contribution and evaluation metric is information/concept leakage. Furthermore, the revised manuscript is now clearer and more accessible, thanks to the inclusion of diagrams and the refined overall layout. I will adjust my score accordingly.
> >
> > However, considering the very slight—or negligible—improvement in prediction accuracy (which I believe should be the primary focus, as suggested by the manuscript’s statements), I recommend that the authors revisit the experimental section to better justify the importance of concept leakage as a contribution.
> >
> > Yours sincerely,

---

> > > ### Author Response · Authors · 2024-11-24
> > > **Reply to further concerns**
> > >
> > > We appreciate the thoughtful engagement of the reviewer with our discussion.
> > >
> > > **Unclear claims.**
> > > It appears there has been a misunderstanding regarding the claims we made and the results shared during the rebuttal. In our manuscript, we articulated our contributions as follows: (i) introducing an Information Bottleneck (IB)-based theoretical framework for CBMs, (ii) demonstrating the information dynamics in CBMs and our enhanced model, CIBMs, noting that CBMs tend to compress information without enhancing expressiveness, and (iii) presenting two IB-based methods within our CIBMs.
> > >
> > > We are not claiming that our work improves the direct prediction capabilities of traditional CBMs. Rather, our goal was to develop information-theoretically grounded CBMs and propose preliminary models that could serve as a foundation for further exploration into principled CBMs.
> > >
> > > Upon revisiting our manuscript for any inconsistencies regarding these claims, we identified and subsequently revised an statement in the abstract. Additionally, in response to your feedback, we have thoroughly reviewed and realigned the experimental section to reflect our objectives accurately, focusing particularly on the issue of concept leakage.
> > >
> > > We are confident that we have addressed the primary concerns raised by the reviewer in this process.
> > >
> > > **Not negligible results.**
> > > Our work focuses on enhancing CBMs to mitigate concept leakage and improve predictive performance without compromising their core strengths of interpretability and intervenability. The improvements we report (2.86% on CUB, 2.33% on AwA, and 7.59% on APY) may appear modest.  However, they still are significant in the context of already strong baseline performances achieved by CBMs alongside providing an improved interpretability and intervenability.

---

> > > > ### Comment · Reviewer_feSc · 2024-11-26
> > > > **Reply to Author(s)**
> > > >
> > > > Thank you for your rebuttal. After carefully reviewing your responses and the feedback from other reviewers (particularly miqn and U1w5), I prefer to keep my score. I recommend that the author(s) improve the manuscript based on all the reviewers’ comments and suggestions.

---

> > > > > ### Author Response · Authors · 2024-11-26
> > > > > **Follow up to the missing baselines**
> > > > >
> > > > > We thank the reviewer for following up the discussion.  However, we will like to highlight the recent changes of the requested baselines, as well as the mixed theoretical and experimental results we present in the paper.
> > > > >
> > > > > As a follow up to the reviewer’s last question about our experiments, we highlight that we added new results since the last reviewer’s update.  We have added our implementation of CEM results and interventions, added expanded corrupted concepts, and now have hard CBMs.
> > > > >
> > > > > In particular, regarding the differences and requested baselines from the R.U1w5, we highlight that we selected the strongest baseline according to the literature.  Even after the reviewer’s new raised points, the main results showed that the soft CBMs perform the best (i.e., 82.7±0.2 accuracy for the soft vs. hard representations achieving 79.5±0.3).  Thus, we highlight that our selection of the experimental setup corresponds to a strong baseline.  While it will be interesting to see other baselines, we are already presenting sufficient ones to evaluate our proposal.

---

> > > > > ### Author Response · Authors · 2024-11-27
> > > > > **New experiments on hard CBMs**
> > > > >
> > > > > We were able to obtain the missing baselines regarding the hard CBMs.  We added them in Table 2 and Fig. 4.  We hope that the reviewer can review their assessment of our proposal based on the final version and based on all the added experiments that significantly improved our original proposal.

---

### Author Response · Authors · 2024-11-21
**General reply to all the reviewers**

We sincerely thank the reviewers for their valuable comments and suggestions. We believe these insights significantly enhance our work, and we deeply appreciate the detailed feedback provided.

We highlight that, as noted by R.GBnn, **this work is the first to explicitly introduce the IB framework into CBMs** to mitigate concept leakage. Our approach provides a general framework that can be implemented using a variety of methods, showcasing its versatility (R.GBnn). We are pleased that the reviewers recognised the identified strengths of our paper, including:
- The **novelty and clarity** of the proposed idea (R.GBnn, R.feSc, R.U1w5, R.xX2U),
- The **simplicity and usability** of the method (R.GBnn),
- The paper being **well-written** and providing sufficient context (R.GBnn, R.miqn),
- The **extensive and multidimensional experimentation** to evaluate our framework (R.feSc, R.GBnn, R.xX2U),
- And the introduction of a **new metric** to quantitatively assess concept set quality (R.feSc, R.GBnn, R.miqn, R.xX2U), which provides a direct and interpretable evaluation method.

We also recognize the concerns raised by the reviewers and appreciate the opportunity to address them. These include the perceived marginal improvement in prediction accuracy, the application of the IB framework potentially limiting novelty, and other suggestions related to implementation details and experimental scope. In our detailed replies, we will address these points by demonstrating:
1. How the adaptation of IB to CBMs introduces innovations that go beyond a direct application of the framework, particularly in addressing concept leakage and improving interpretability.
2. Why improvements in accuracy are modest but balanced by significant gains in concept quality and intervenability, aligning with the broader goals of explainable AI.
3. How $\text{CIBM}_B$ and $\text{CIBM}_E$ provide complementary strengths, with $\text{CIBM}_B$ excelling in scenarios with frozen encoders and $\text{CIBM}_E$ offering more precise control in dynamic training settings.
4. Our ongoing efforts to repeat experiments for robustness, include comparisons with CEM, and add baseline intervention curves for CBMs to address feedback comprehensively.
5. We are also working to incorporate more implementation details (e.g., estimator design, concept GTs), perform additional runs where feasible, and clarify computational overhead and hyperparameter search in the appendix. These updates aim to strengthen the manuscript, and we will notify reviewers of the changes as they are incorporated.

We incorporated most of the suggested changes into the manuscript. We will ensure to notify the reviewers when the missing ones are integrated. We hope these revisions, alongside our detailed responses to individual comments, will address the concerns and enhance the clarity and impact of our work.

---

> ### Author Response · Authors · 2024-11-26
> **Follow up to the requested experiments**
>
> In summary, after the exchanges with the reviewers, and after our previous update, we were able to perform the following experiments and added them to the paper:
>
> 1. Inclusion of CEM results across multiple datasets (AwA2 and APY) (Table 2).
> 2. Inclusion of CEM intervention results on both CUB and AwA2 (Figure 4).
> 3. Addition of hard-concept CBM (CBM HJ in Figure C.1).
> 4. Expansion of the analysis on corrupted concepts (Figure C.2) to evaluate robustness and leakage mitigation.
>
> Our evaluation of CEM corresponds to the basic setup since we couldn’t implement the full version during the rebuttal time.  Nevertheless, our implementation matches the one reported from the authors, but the basic version doesn’t perform well in the interventions.
>
> Moreover, we give individual replies to the reviewers comments in their respective threads.

---

> ### Author Response · Authors · 2024-11-27
> **New experiments for hard and soft CBMs**
>
> As a final update about the requested experiments, we were able to perform the experiments regarding **hard and soft CBMs trained jointly and independently** in CUB and AwA2 datasets.  We added those results into the overall performance in Table 2, as well as the intervention results in Fig. 4.
>
> We highlight that our results show better performance in comparison to the hard CBMs.  In the interventions evaluations, there is a bump in performance in the hard CBMs due to their nature, but they suffer in coarser datasets.  We added a discussion about the differences of these training setups and the proposed CIBMs in Appendix D.

---

### Meta-Review · Area_Chair_Pvyv · 2024-12-20

**Metareview:**

The paper takes a step towards integrating the IB framework into CBMs and targets a meaningful problem—concept leakage in interpretable models.

However, after the rebuttal and additions, doubts remained regarding whether the paper provides sufficiently strong and comprehensive empirical evidence to justify its claims, especially compared to existing leakage mitigation strategies and well-tuned baselines. While the authors made commendable efforts, the consensus among the reviewers was that the paper could be further strengthened.

Therefore, for the benefit of this paper, we regretfully reject it for now. We encourage the authors to incorporate more comparisons, clearer demonstrations of leakage mitigation, and possibly refined metrics or analyses for a future submission.

**Additional Comments On Reviewer Discussion:**

In the review proces, the reviewers have raised some major conerns as follows:


1. **Improvements in performance and leakage reduction not sufficiently convincing, especially with stronger baselines and best practices:**
   - *Reviewer U1w5*: Explicitly requested comparisons against better-tuned CBMs (e.g., hard concept representations and independent training) and found the improvements unconvincing without these baselines.
   - *Reviewer feSc*: Expressed that the observed accuracy gains were modest and requested stronger justification.
   - *Reviewer miqn*: Noted the need for more comprehensive comparisons and was not fully convinced by the experimental support for improvements.

2. **Technical Novelty of the method questioned (method builds on existing IB techniques):**
   - *Reviewer GBnn*: Pointed out that the technical method for minimizing MI relied on existing methods and wanted more discussion on why the chosen estimator was preferable.
   - *Reviewer U1w5*: Implied that the approach was not providing fundamentally new algorithmic insights beyond the application of IB concepts.

3. **Requests for additional comparisons with recognized leakage-reduction methods and more thorough empirical analyses across multiple datasets and baselines:**
   - *Reviewer U1w5*: Repeatedly asked for comparisons with hard concept CBMs, independent training schemes, and other established models like CEM on all datasets.
   - *Reviewer feSc*: Encouraged further justification through broader evaluations.
   - *Reviewer miqn*: Suggested that adding more baselines and demonstrating the full potential of the idea would strengthen the paper.



During Review-AC discussion, most reviewers acknowledge that these concerns have not been sufficiently solved. Note that this is not a disencouragement, and we believe this paper should be a strong submission after address these concerns.

---

### Decision · Program_Chairs · 2025-01-22

Reject